# CHARTNEXUS: EVALUATING MULTI-CHART REASONING CAPABILITIES OF MULTIMODAL LARGE LANGUAGE MODELS

## ABSTRACT

While Multimodal Large Language Models (MLLMs) have achieved remarkable success on single-chart question-answering tasks, reaching over 90% accuracy on benchmarks such as PlotQA, this apparent success masks a critical limitation. Current models struggle to perform well on complex, multi-chart reasoning tasks that closely mirror real-world analytical scenarios. In professional document analysis, users typically integrate information across multiple visualizations within rich contextual frameworks, rather than examining isolated charts, a capability that remains largely unexplored in existing evaluations. To bridge this gap, we introduce ChartNexus, a novel and challenging benchmark specifically designed to assess multi-chart reasoning capabilities of MLLMs in authentic document contexts. ChartNexus comprises 1,370 carefully curated question-answering pairs derived from 6,793 real-world charts spanning 18 domains, including scientific papers, government reports, and industry analyses. Each question demands complex reasoning skills, such as comparative analysis, sequential information integration, and cross-modal synthesis between visual and textual elements. We design a comprehensive taxonomy featuring 4 high-level difficulty categories and 11 fine-grained sub-categories to systematically evaluate these capabilities. Our comprehensive evaluation of 23 state-of-the-art MLLMs reveals significant performance degradation compared to single-chart benchmarks. While the best commercial model achieves over 90% accuracy on simpler tasks, its performance drops by more than half on ChartNexus. Through systematic failure analysis, we identify critical weaknesses in current models' ability to maintain working memory across multiple charts, perform cross-modal reasoning, and integrate contextual information effectively. ChartNexus establishes a new frontier for evaluating complex chart understanding capabilities, demonstrating that robust multi-chart reasoning remains an open challenge. Our benchmark and comprehensive analysis provide the research community with essential diagnostic tools to advance the development of more capable and practically useful MLLMs for real-world document analysis scenarios.

## 1 INTRODUCTION

Data visualization, especially charts, serves as a fundamental medium for conveying complex information across scientific research, financial reporting, and journalism (Huang et al., 2025). The rapid development of MLLMs has brought unprecedented opportunities for automating the understanding of these visual representations. Chart Question-Answering (ChartQA) has emerged as a critical benchmark task that evaluates how well these models can integrate visual perception with cognitive reasoning. The field has witnessed a remarkable paradigm shift from specialized domain-specific models (Methani et al., 2020) to large-scale foundation models like GPT-4o, has driven significant progress in ChartQA.

However, this apparent success masks significant limitations in current MLLM capabilities. Leading MLLMs are approaching or surpassing human-level performance on established

Table 1: Comparison with other benchmarks

| Dataset | Real-World Charts | Human Annotated | Multi Charts | Chart Types | Task Types | Unanswerable Question | Fine-Grained Difficulty | Multilingual | Document Context |
|---|---|---|---|---|---|---|---|---|---|
| PlotQA (Methani et al., 2020) | ✓ | ✓ | ✗ | 3 | 3 | ✗ | ✗ | ✗ | ✗ |
| ChartQA (Masry et al., 2022) | ✓ | ✓ | ✗ | 3 | 4 | ✗ | ✗ | ✗ | ✗ |
| RealCQA (Ahmed et al., 2023) | ✓ | ✗ | ✗ | 5 | 4 | ✗ | ✗ | ✗ | ✗ |
| ChartLlama (Han et al., 2023) | ✗ | ✗ | ✗ | 10 | 7 | ✗ | ✗ | ✗ | ✗ |
| UniChart (Masry et al., 2023) | ✓ | ✗ | ✗ | 3 | 4 | ✗ | ✗ | ✗ | ✗ |
| ChartBench (Xu et al., 2023) | ✗ | ✗ | ✗ | 9 | 5 | ✗ | ✗ | ✗ | ✗ |
| ChartSFT (Meng et al., 2024) | ✓ | ✗ | ✗ | 4 | 5 | ✗ | ✗ | ✗ | ✗ |
| SBS_figures (Shinoda et al., 2024) | ✗ | ✗ | ✗ | 10 | 11 | ✗ | ✗ | ✗ | ✗ |
| Dcqa (Wu et al., 2023) | ✓ | ✓ | ✗ | 6 | 2 | ✗ | ✓ | ✗ | ✗ |
| Chart-llm (Ko et al., 2024) | ✓ | ✗ | ✗ | 10 | 4 | ✗ | ✓ | ✗ | ✓ |
| MultiChartQA (Zhu et al., 2025c) | ✓ | ✓ | ✓ | - | 4 | ✗ | ✗ | ✗ | ✗ |
| ReachQA (He et al., 2024b) | ✗ | ✗ | ✗ | 10 | 2 | ✗ | ✗ | ✗ | ✗ |
| ChartInsights (Wu et al., 2024) | ✓ | ✗ | ✗ | 7 | 10 | ✗ | ✗ | ✗ | ✗ |
| RealCQA-V2 (Ahmed et al., 2024) | ✓ | ✗ | ✗ | 5 | 3 | ✗ | ✗ | ✗ | ✗ |
| StructChart (Xia et al., 2023) | ✗ | ✗ | ✗ | 3 | 3 | ✗ | ✗ | ✗ | ✗ |
| CharXiv (Wang et al., 2024b) | ✓ | ✓ | ✓ | 15 | 6 | ✓ | ✗ | ✗ | ✗ |
| DomainCQA (Zhong et al., 2025) | ✓ | ✗ | ✓ | - | 6 | ✗ | ✓ | ✗ | ✗ |
| ChartQA-MLLM (Zeng et al., 2025) | ✗ | ✓ | ✗ | 11 | 4 | ✗ | ✗ | ✗ | ✗ |
| SPIQA (Pramanick et al., 2024) | ✓ | ✗ | ✓ | - | 3 | ✗ | ✗ | ✗ | ✓ |
| ChartX (Xia et al., 2024) | ✗ | ✗ | ✗ | 18 | 7 | ✗ | ✗ | ✗ | ✗ |
| PolyChartQA (Xu et al., 2025a) | ✓ | ✗ | ✗ | 16 | - | ✗ | ✗ | ✓ | ✗ |
| ChartQAPro (Masry et al., 2025a) | ✓ | ✓ | ✓ | 9 | 5 | ✓ | ✗ | ✗ | ✓ |
| ChartNexus (Ours) | ✓ | ✓ | ✓ | 17 | 6 | ✓ | ✓ | ✓ | ✓ |

benchmarks such as FigureQA (Kahou et al., 2017), UniChart (Masry et al., 2023). Yet recent evaluations on more challenging single-chart benchmarks, like ChartQAPro (Masry et al., 2025a), DomainCQA (Zhong et al., 2025), reveal substantial performance drops when models encounter diverse visual elements and complex question types. This performance degradation indicates that existing benchmarks lack sufficient complexity to adequately assess model capabilities in realistic chart understanding scenarios.

More critically, a fundamental dimension of chart understanding remains underexplored: multi-chart reasoning. In real-world analytical workflows, users rarely examine charts in isolation. Instead, they must integrate information across multiple visualizations, often combining insights with the surrounding textual context to form a comprehensive understanding. This process demands cross-modal reasoning and multi-hop inference across diverse information sources. Despite its importance in practical applications, most existing benchmarks are confined to single-chart scenarios. Although MultiChartQA (Zhu et al., 2025c) has begun to address this gap, its coverage of diverse chart domains and the complexity of its reasoning chains remain limited and focused only on charts themselves. The research community urgently requires larger, more complex benchmarks with broader real-world scenarios and more extensive reasoning capabilities.

Moving from single-chart to multi-chart QA constitutes a qualitative leap in computational requirements, representing far more than a simple incremental increase in difficulty. Single-chart tasks assess a model's ability to parse visual elements within confined contexts, such as identifying peak values in line graphs, extracting specific data points, or performing straightforward calculations. The analytical scope remains strictly bounded within individual images. Multi-chart QA, particularly requiring multi-hop and comparative reasoning, demands fundamentally different model capabilities. Models must retain information extracted from one chart while processing subsequent visualizations, compare attributes across various visual contexts, and track entities as they evolve across multiple representations. This requires models to manage larger information spaces while executing multi-step inferences across interconnected visual elements.

Therefore, we introduce **ChartNexus**, a novel, challenging benchmark designed to assess the multi-chart reasoning capabilities of MLLMs in authentic document contexts. Chart-Nexus comprises 6,793 carefully selected charts from real-world documents, including scientific papers, government reports, and industry analyses, and features 1,370 high-quality human-annotated QA pairs. Each question demands complex reasoning skills, such as comparative analysis across multiple charts and cross-modal synthesis between visual elements and their surrounding text. We design a comprehensive taxonomy to evaluate these capabilities, featuring 4 high-level difficulty categories and 11 fine-grained subcategories. Our

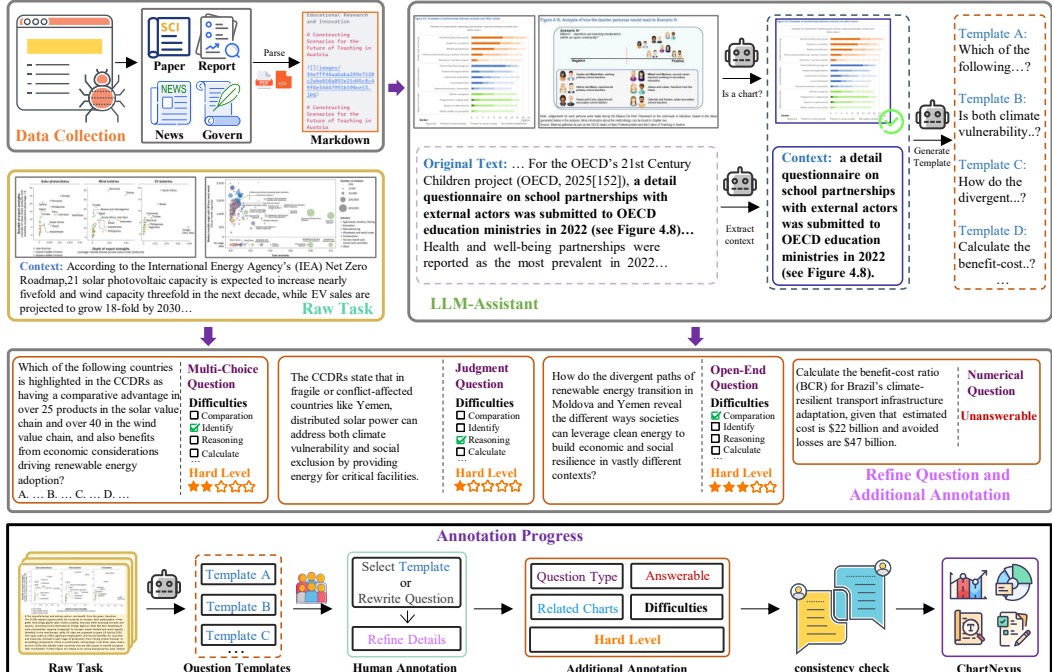

Figure 1: Overview of data construction. We first collect a diverse range of documents from the internet. Then, employ MLLMs to filter the raw data and generate several candidate question templates. Following this, human annotators select the most suitable template, refine the question, and complete the annotation.

comprehensive evaluation of 23 leading MLLMs reveals significant performance degradation compared to single-chart benchmarks. While the best-performing model achieves over 90% accuracy on simple tasks, its score drops by more than half on ChartNexus. Through systematic failure analysis, we identify critical weaknesses in the current MLLMs' ability to maintain information across multiple charts, perform cross-modal reasoning, and effectively integrate contextual information. Our main contributions are as follows.

- We introduce the ChartNexus benchmark, a novel and highly challenging multi-chart QA benchmark featuring charts from authentic real-world documents, human-annotated question-answer pairs, and associated descriptive text, designed to rigorously test complex cross-modal synthesis and reasoning abilities.

- We comprehensively evaluate leading closed- and open-source MLLMs, establishing realistic performance that reveals current models' true capabilities and limitations.

- We provide detailed failure analysis that moves beyond simple accuracy metrics to offer a systematic taxonomy of failure modes, delivering insights into why and how current models struggle with multi-chart reasoning and illuminating directions for future research.

## 2 RELATED WORKS

**Existing Benchmarks.** Early studies lay the foundation for the field, but their data relies on synthetic charts, creating a significant gap with the real world. FigureQA provides over a million QA pairs based on synthetic, scientific-style charts (Kahou et al., 2017). It establishes the task paradigm with templated questions (e.g., identifying max/min values), but its synthetic nature lacks the diversity of real data. Although PlotQA uses charts scraped from the web, ensuring authentic chart styles, its QA pairs are similarly constrained by templates (Methani et al., 2020). ChartQA utilizes the T5 model for auxiliary generation, which improves issues such as unnatural text, resulting in the generation of template-based questions (Masry et al., 2022). Recently, PolyChartQA (Xu et al., 2025a) has expanded the field's horizon by introducing a large-scale multilingual benchmark. However, like its prede-

Table 2: Chart types in ChartNexus

| Bar | Line | Pie | Table | Scatter | Tree | Radar | Area | Other | Sunburst | Graph | Boxplot | Sankey | Heatmap | 3D | Candlestick | Funnel |
|------|------|-----|-------|---------|------|-------|------|-------|----------|-------|---------|--------|---------|-----|-------------|--------|
| 2704 | 1947 | 330 | 37 | 261 | 243 | 20 | 270 | 774 | 17 | 15 | 51 | 19 | 45 | 36 | 11 | 13 |

Table 3: Sub-categories of fine-grained difficulties in ChartNexus

| Numerical | Identification | | | | | Comparison | | | Reasoning | |
|-----------|---------|-------|-------|---------|---------|-----------|-------|----------------|---------------|-------------------|
| calculate | element | color | shape | overlap | 3d-chart | numerical | trend | a lot of charts | chart context | general knowledge |
| 415 | 132 | 127 | 145 | 133 | 125 | 63 | 519 | 67 | 516 | 393 |

cessors, it remains constrained to single-chart scenarios and does not address the complexity of cross-chart information synthesis. These datasets contain vast amounts of data, but the quality of their QA pairs is limited to templates, simple data retrieval, and fixed-vocabulary questions. While these QA pairs include out-of-vocabulary words, which are challenging for models of this era, they are no longer sufficient for evaluating modern MLLMs.

**Challenges in Single-Chart Understanding.** Given the limitations of these benchmarks, recent research has begun to introduce new dimensions of difficulty into single-chart tasks. ChartLLama uses GPT-4 to construct its tasks. Compelling models to possess advanced chart understanding and code-based plotting abilities to achieve higher scores through new tasks like chart reconstruction, generation, and editing (Han et al., 2023). ChartQAPro aims to address the lack of diversity in ChartQA (Masry et al., 2025a) by introducing more complex visual forms such as info-graphics and dashboards, as well as more challenging question types such as conversational, hypothetical, and unanswerable questions. Other benchmarks such as UniChart, MatCha, and ChartAssistant have introduced open-ended questions like inverse-rendering charts into code or tables to test models' deeper understanding of charts (Masry et al., 2023; Liu et al., 2023; Meng et al., 2024).

However, these datasets are limited to understanding single charts. Furthermore, many are annotated using LLMs, making the quality highly dependent on the prompts, creating a significant gap with the needs of professional researchers in real-world chart analysis.

**Multi-Chart QA.** Beyond single charts, multi-chart QA has recently become a new research hotspot. MultiChartQA crawls charts from websites and features manually annotated questions that test various reasoning abilities of models (Zhu et al., 2025c). SPIQA focuses on scientific charts from top-tier computer science conference papers and uses Gemini to generate candidate questions, which are then refined by humans (Pramanick et al., 2024).

These excellent studies have extended chart QA from single- to multi-chart scenarios, significantly raising the requirements for models' visual reasoning capabilities. However, in real-world scenarios, analysts rarely draw conclusions based on just a few charts alone. Charts are often used as a visualization method to help personnel understand the content of the document more quickly. A deep understanding of charts is inseparable from the specific descriptions provided by their surrounding context. While only simple numerical values and trend information can be obtained from the chart itself, the deeper causal factors are hidden in the contextual text associated with that chart. In real-world document QA scenarios, MLLMs may produce incorrect answers by focusing only on the chart and overlooking crucial information within the surrounding text. For an illustrative example of this failure mode, please refer to the error case Figure 35 in Appendix F.

Due to the lack of benchmarks for multi-chart reasoning that incorporate contextual information, we introduce ChartNexus to effectively evaluate the multi-modal reasoning capabilities of existing models. ChartNexus not only incorporates the pursuit of authenticity, diversity, and complex reasoning from ChartQAPro but also introduces the novel multi-chart reasoning dimension pioneered by MultiChartQA, along with the innovative inclusion of cross-modal reasoning with document context. Through these comprehensive features, ChartNexus establishes a new frontier specifically designed to challenge MLLMs.

## 3 Construction of ChartNexus

ChartNexus is a benchmark designed to reflect real-world document chart comprehension needs, comprising a total of 6,793 charts and 1,370 question-answering (QA) tasks. All

charts are sourced from real-world documents and span various types of documents and topics. This section details the design principles of the ChartNexus benchmark, its data construction process, the QA annotation methodology, and data analysis of ChartNexus. Our data construction pipeline is illustrated in Figure 1.

## 3.1 Data Collection

The primary motivation behind constructing ChartNexus is to establish a benchmark that genuinely reflects the cognitive processes involved in analyzing multi-chart documents in real-world scenarios. We collect recent source documents that contain substantive information from real-world, data-intensive websites. This approach ensures that the charts and their semantic relationships are authentic and require reasoning, thus simulating a real-world application while avoiding overlap with the training corpora of existing models as much as possible. Specifically, we collect data, including charts and their relevant contextual information, from 10 distinct data sources.

**Scientific Papers from arXiv:** Referencing the work of SPIQA (Pramanick et al., 2024), which collects documents from top-tier computer science conference papers and provides all charts along with their descriptions, we select 425 source documents and re-annotate QA pairs to meet our requirements.

**In-depth News Reports:** We obtain news reports from Statista and the Pew Research Center. While each article from the Pew Research Center contains multiple charts, reports from Statista typically include a single chart. To construct multi-chart reasoning tasks, we search for additional reports on the same topics within Statista and group them to create multi-chart QA entries. Ultimately, we acquire 318 and 334 data entries from Statista and Pew Research Center, respectively.

**Government Reports:** This category includes reports from the National Bureau of Statistics of China (1,000 entries), the Guizhou Provincial Statistical Bulletin (17 entries), the World Bank (300 entries), and the Organisation for Economic Co-operation and Development (OECD) (282 entries). We download statistical data and research reports, from which we extract charts and their related contexts.

**Industry Data:** We also collect research reports from specific industries, including the China Internet Network Information Center (CNNIC), Communications World, and the National Consortium for the Study of Terrorism and Responses to Terrorism (START). These reports contain research documents on specialized fields such as the internet, telecommunications, and public safety. We create QA pairs from these sources to investigate the visual-textual understanding capabilities of MLLMs in professional domains.

Table 4: ChartNexus dataset statistics. Tokens are calculated based on the Qwen3 tokenizer.

| Statistics | Value |
|---|---|
| **Charts** | |
| total charts | 3198 |
| *Sub-Charts* | |
| - max | 57 |
| - mean | 4.78 |
| *Related Charts Per Question* | |
| - max | 7 |
| - mean | 3.67 |
| **Average Tokens** | |
| context | 95.71 |
| question | 66.64 |
| answer | 125.86 |
| **Answer Type** | |
| Multi Choice | 335 |
| Judge | 200 |
| Numerical | 276 |
| Open-Ended (vocabulary) | 187 |
| Open-Ended (sentence) | 263 |
| Unanswerable | 109 |

The data collected from these sources are primarily in PDF or HTML format. For PDF documents, we use MinerU for parsing, converting the text into Markdown, and segmenting charts and tables as images (Wang et al., 2024a; He et al., 2024a). For HTML files, we extract the main body of the text and chart links, saving the content and images locally. While HTML data can be directly converted into a structured document based on its tags, for Markdown data, we parse its syntax, using headings to define the nesting hierarchy, and then convert it to a structured JSON document. It is noteworthy that the initially extracted

images were not all charts. Therefore, we employ Qwen2.5-VL-7B for a preliminary filter, retaining only those images identified as charts.

## 3.2 Question-Answer Annotation

A core design principle of ChartNexus is that each question must necessitate multi-hop reasoning, compelling a model to synthesize information from at least two charts. To ensure high-quality and complex QA pairs, we employ a human-in-the-loop annotation pipeline that uses an LLM to assist expert annotators, and iteratively refine the annotation process and guidelines, as shown in Figure 1. Before beginning the formal annotation, we first invite graduate students with backgrounds in data analysis and deep learning to conduct a pilot study. Through this process, we finalize the necessary annotation items for the benchmark and provide the LLM in our formal pipeline with the few-shot examples needed to generate candidate questions. Trained annotators then either refine these suggestions or create entirely new questions to ensure they are logically sound, deeply integrated with the provided charts, and require non-trivial reasoning. Crucially, annotators also provide ground-truth answers, with a portion of questions intentionally designed to be unanswerable from the given context to test model robustness.

To validate the quality and consistency of our dataset, we conducted a rigorous verification process. A randomly selected 20% subset of the annotations was independently re-annotated, and we achieved an inter-annotator agreement rate of 93.4%. This high consistency underscores the clarity of our annotation guidelines and the objective nature of the tasks. A final expert review resolved any discrepancies to establish the definitive ground truth. More details on the annotation pipeline, including the pilot study, question generation prompts, and annotator guidelines, are available in the Appendix A.

## 3.3 Data Analysis

ChartNexus contains 17 types of charts and tables from 3,198 original real-world documents, with bar charts accounting for 39.8%, line charts for 28.7%, pie charts for 3.44%, and the remaining 14 types (such as scatter plots, area charts, etc.) shown in Table 2. Furthermore, 16.69% charts that contain subplots, with an average of 4.78 subplots per chart. This diversity evaluate models' capabilities of processing global complex layouts and handling local information. On average, each context related to the charts contains 95.71 tokens, which brings the challenge of carrying text and vision together.

The distribution of topics about our charts is presented in Figure 2. The charts span 18 different domains, ensuring both breadth and depth. Economics is the most dominant subject. This is followed by Social and Government, which typically involves the analysis of complex socioeconomic data. Furthermore, ChartNexus also covers a wide array of specialized fields, including Science, Finance, as well as environment, education, etc. On average, each task involves 1.65 subject domains. This indicates that many questions require models to perform comprehensive analysis by integrating background knowledge from different fields, which aligns with the interdisciplinary nature of real-world problems.

Our ChartNexus dataset contains question-answer pairs in both English and Chinese, with questions averaging 66.64 tokens and answers averaging 125.86 tokens in length. On average, each question requires information from 3.72 charts to be answered. ChartNexus has 4 types of questions and 6 types of answer formats. The primary formats include Open-Ended question and Multi-Choice questions. There are 8% questions that are intentionally designed to be unanswerable. To more precisely evaluate specific model capabilities, we classify the task difficulties into 11 fine-grained categories (see Table 3).

## 4 Experiments

To comprehensively evaluate the capabilities of MLLMs in ChartNexus, we conduct a series of experiments. This section details our experimental setup, presents the overall performance

Table 5: Performance of MLLMs on ChartNexus. We report the Accuracy (%) and F1 score calculated from SEAT method (Zhu et al., 2025b). Bold values indicate the best result within each category.

| Model | Question Type | | | | | | Difficulty | | | | Language | |
|---|---|---|---|---|---|---|---|---|---|---|---|---|
| | Multi Choice | Judge | Approximate Value | Open-Ended (vocabulary) | Open-Ended (sentence) | Unanswerable | Numerical | Identify | Compare | Reason | ZH | EN |
| *Commercial Model* | | | | | | | | | | | | |
| GPT-4o | 58.62 | 67.56 | **41.37** | 44.43 | 74.13 | 23.80 | 65.60 | 63.63 | **47.82** | 66.46 | 70.58 | 62.61 |
| GPT-o4-mini | 62.06 | 60.81 | 38.70 | 44.45 | 81.71 | 16.67 | **68.13** | 63.44 | 43.47 | **69.34** | 77.94 | 63.89 |
| GPT-o3 | 63.79 | 59.45 | 21.87 | 40.74 | **83.42** | 19.04 | 66.67 | 61.37 | 42.23 | 68.84 | **80.88** | 61.84 |
| Claude-Sonnet4 | **65.71** | 70.96 | 32.22 | 40.05 | 72.66 | 18.19 | 63.60 | 60.15 | 45.43 | 67.87 | 79.41 | 61.25 |
| Gemini-2.5-Pro | 56.89 | 60.81 | 15.62 | 31.48 | 80.00 | 40.47 | 61.94 | 64.13 | 39.13 | 63.50 | 72.05 | 58.76 |
| Gemini-2.5-Flash | 55.17 | 54.05 | 28.12 | 35.18 | 71.26 | **50.03** | 56.78 | 65.97 | 34.78 | 57.14 | 57.35 | 57.09 |
| Doubao-Seed-1.6 | 46.55 | 43.24 | 37.50 | 23.37 | 70.85 | 45.23 | 53.77 | 59.31 | 21.73 | 56.97 | 67.64 | 49.23 |
| Qwen-VL-MAX | 62.06 | **75.67** | 30.02 | **50.00** | 71.42 | 26.19 | 65.18 | **67.58** | 34.78 | 67.55 | 65.70 | **64.58** |
| HunYuan-Turbos-Vision | 59.64 | 70.27 | 29.03 | 29.62 | 74.85 | 11.90 | 63.60 | 57.63 | 30.43 | 67.46 | 76.11 | 58.95 |
| HunYuan-Vision | 61.14 | 59.45 | 19.53 | 20.37 | 61.14 | 16.68 | 50.15 | 43.05 | 26.08 | 55.05 | 60.29 | 48.14 |
| Ernie-4.5-Turbo-VL | 51.72 | 52.05 | 31.30 | 30.18 | 65.71 | 45.23 | 54.25 | 54.48 | 26.08 | 57.14 | 62.68 | 51.54 |
| *Open-Source Model* | | | | | | | | | | | | |
| SmolVLM-2.3B | 8.62 | 10.81 | 6.25 | 1.88 | 1.14 | 26.19 | 2.83 | 10.34 | 4.34 | 3.86 | 2.98 | 4.93 |
| Phi-4-multimodal-Instruct | 35.08 | 55.40 | 12.5 | 18.51 | 31.42 | 9.52 | 31.86 | 33.10 | 13.04 | 36.60 | 20.59 | 35.80 |
| Bagel | 29.31 | 33.78 | 15.62 | 24.07 | 41.14 | 38.09 | 33.64 | 26.89 | 17.39 | 36.49 | 48.52 | 30.46 |
| Kimi-VL-A3B-Thinking | 53.44 | **67.54** | 25.00 | 29.62 | 72.83 | 21.42 | 58.75 | **56.25** | **43.47** | **64.88** | **73.13** | 56.17 |
| Qwen2.5-VL-7B | 34.48 | 31.08 | 35.02 | 18.51 | 46.67 | **54.76** | 41.13 | 23.44 | 21.73 | 43.54 | 42.43 | 44.92 |
| GLM-4.1V-9B | 50.03 | 49.31 | **35.61** | **33.32** | 50.28 | 35.71 | 53.02 | 39.31 | 37.73 | 54.33 | 63.41 | **59.23** |
| InternVL3-14B | 57.89 | 48.49 | 26.25 | 18.51 | 72.21 | 23.80 | 52.54 | 50.17 | 30.43 | 54.58 | 65.14 | 48.79 |
| Qwen2.5-VL-32B | 59.65 | 56.02 | 32.50 | 20.75 | 63.36 | 38.09 | 56.06 | 52.55 | 38.66 | 47.84 | 62.90 | 55.24 |
| InternVL3-38B | **60.34** | 55.56 | 31.25 | 30.18 | **74.28** | 28.57 | **59.62** | 55.94 | 39.13 | 61.72 | 71.64 | 56.65 |
| *Chart Model* | | | | | | | | | | | | |
| ChartGemma | 6.89 | **21.62** | 3.52 | **11.53** | 2.87 | 21.42 | 6.30 | 6.94 | **13.04** | 5.68 | 7.35 | 8.07 |
| ChartInstruct-LLama2 | 24.13 | 19.17 | 6.25 | 5.56 | 9.19 | 33.34 | 12.65 | 15.17 | 8.69 | 12.50 | 9.09 | 13.23 |
| ChartMoe | **41.37** | 20.27 | **12.52** | 7.40 | **24.57** | **47.61** | **21.69** | **23.44** | 10.27 | **25.22** | **30.88** | **21.23** |

of various models, and provides an in-depth analysis of their strengths and weaknesses across different tasks, difficulties, and languages.

## 4.1 EXPERIMENTAL SETUP

**Model Selection.** We select a series of MLLMs that represent the state-of-the-art performance to ensure a comprehensive and impartial evaluation of the field. Our selection encompasses the latest commercial models and leading open-source models with varying parameter scales. For commercial models, we primarily focus on the series from OpenAI, Anthropic, and Google. For open-source models, our main choices include the Qwen and InternVL series (Bai et al., 2025; Zhu et al., 2025a), as well as several specialized models designed for chart-related tasks (Xu et al., 2025b; Masry et al., 2024; 2025b).

**Setup.** To ensure the reproducibility of our experiments, we follow the official guidelines to call the APIs when testing the commercial models. For the open-source models, we adapt our benchmark with minimal modifications to the example code provided in each model's repository and conducted the experiments with NVIDIA RTX 6000 Ada GPUs.

**Evaluation Metric.** We employ scoring methods for different types of questions. For "Multiple-Choice", "Judgement", "Open-Ended vocabulary" questions and "Unanswerable" questions, we report the model's performance using accuracy. Since many answers contain variations, such as different numerical units, that make traditional character-matching methods ineffective, we employ a Qwen3-32B model as an automated evaluator to judge the correctness of the answers. For questions of the "Approximate Value" type (e.g., values estimated from charts), we consider an answer to be correct if the model's estimation fell within a 5% margin of error relative to the ground truth. For "Open-ended sentence" questions, we utilize the SEAT method (Zhu et al., 2025b) to calculate the F1 score. Specifically, this method involves decomposing the question and ground-truth answer into multiple sub-questions and corresponding sub-answers. The F1 score is then computed based on the matching between the model's generated response and these sub-answers.

## 4.2 RESULTS

**Human Performance Upper Bound.** To establish a rigorous upper bound, we recruited experts to evaluate a sample of the ChartNexus (detailed in Appendix H.1). Humans achieved an average accuracy of 93.3% on Boolean and Vocabulary tasks, and 85.7% on complex Open-Ended Sentence tasks. This highlights a significant gap compared to SOTA

Table 6: Performance of MLLMs on ChartNexus using Chain-of-Thought strategy.

| Model | Question Type | | | | | | Difficulty | | | | Language | |
|---|---|---|---|---|---|---|---|---|---|---|---|---|
| | Multi Choice | Judge | Approximate Value | Open-Ended (vocabulary) | Open-Ended (sentence) | Unanswerable | Numerical | Identify | Compare | Reason | ZH | EN |
| *Commercial Model* | | | | | | | | | | | | |
| GPT-4o | 65.57 | 74.29 | 53.71 | 38.56 | 72.88 | 22.31 | 64.61 | 66.29 | 46.34 | 65.12 | 66.30 | 62.82 |
| Claude-Sonnet4 | 67.47 | 72.54 | 45.82 | 42.47 | 70.25 | 19.24 | 67.47 | 62.15 | 43.74 | 63.45 | 75.54 | 63.23 |
| Gemini-2.5-Pro | 62.50 | 66.97 | 31.02 | 36.88 | 83.20 | 45.59 | 67.32 | 69.33 | 44.25 | 68.82 | 76.88 | 64.14 |
| Doubao-Seed-1.6 | 47.48 | 42.80 | 38.65 | 23.15 | 71.48 | 46.33 | 54.20 | 58.23 | 22.85 | 63.54 | 68.68 | 48.63 |
| Qwen-VL-MAX | 67.82 | 79.25 | 35.58 | 53.25 | 76.13 | 30.71 | 63.45 | 72.10 | 39.66 | 64.14 | 66.25 | 67.32 |
| *Open-Source Model* | | | | | | | | | | | | |
| Qwen2.5-VL-7B | 47.61 | 50.20 | 40.33 | 17.30 | 49.49 | 44.29 | 46.59 | 28.57 | 29.26 | 48.72 | 43.71 | 47.95 |
| GLM-4.1V-9B | 62.16 | 61.14 | 45.22 | 30.09 | 68.77 | 17.27 | 67.34 | 56.09 | 43.90 | 56.44 | 68.76 | 58.03 |
| InternVL3-14B | 56.47 | 60.65 | 36.56 | 25.29 | 56.25 | 14.37 | 63.02 | 48.78 | 41.46 | 62.08 | 58.94 | 51.12 |
| Qwen2.5-VL-32B | 67.85 | 62.63 | 46.60 | 31.37 | 72.67 | 22.50 | 65.77 | 60.52 | 41.46 | 58.08 | 64.58 | 58.37 |
| InternVL3-38B | 52.09 | 71.42 | 58.88 | 22.03 | 58.96 | 20.68 | 64.98 | 51.21 | 53.84 | 63.83 | 64.13 | 63.08 |
| *Chart Model* | | | | | | | | | | | | |
| ChartGemma | 13.16 | 17.70 | 9.16 | 9.74 | 3.41 | 44.51 | 10.17 | 13.25 | 8.10 | 9.16 | 9.36 | 10.82 |
| ChartInstruct-LLama2 | 20.57 | 27.00 | 8.47 | 5.56 | 6.45 | 32.29 | 14.89 | 18.32 | 9.75 | 13.73 | 14.97 | 14.15 |
| ChartMoe | 61.31 | 19.32 | 13.33 | 11.61 | 26.44 | 48.66 | 29.06 | 17.78 | 14.63 | 31.66 | 26.67 | 31.06 |

models, which hover around 60-70%, confirming that ChartNexus remains a challenging benchmark for current MLLMs.

**Main Results.** Commercial models demonstrate superior overall performance. Models like the ChatGPT family and Qwen-VL-MAX achieve the highest scores across most categories. For example, GPT-o3 shows strong performance in generating open-ended sentences (83. 42%) and handling queries in Chinese (80. 88%). Open-source models exhibit significant performance variability. While larger models such as InternVL3-38B and Kimi-VL-A3B-Thinking are competitive, many smaller models struggle significantly. Models like SmolVLM-2.3B and Phi-4-multimodal-Instruct post scores below 5% in some categories, highlighting that strong multi-chart reasoning has not yet been democratized in smaller, more accessible models. A surprising finding is the underperformance of specialized chart models. ChartGemma, ChartInstruct-Llama2, and ChartMoe all lag considerably behind the leading general-purpose commercial and open-source MLLMs. This suggests that their specialized training has not been sufficient to overcome the complex, multi-step reasoning required by this benchmark.

**Performance by Task and Difficulty.** Most models perform best on generating open-ended sentences, where they can formulate descriptive answers. In contrast, they are weakest on tasks requiring precise numerical approximation and identifying unanswerable questions. The difficulty with numerical tasks points to a known weakness in MLLMs for precise calculation. Tasks that require estimation and the inability to correctly identify unanswerable questions indicate a tendency to hallucinate or force an answer from the provided charts. Across the board, models find identification and trend analysis to be easier than tasks requiring deeper reasoning. Performance drops significantly for comparison tasks, which often require integrating information from multiple charts or performing multi-hop logical steps. This underscores that complex reasoning remains a primary challenge for all models. Many leading models perform better in Chinese than in English. This is especially true for models developed in China, such as HunYuan and Kimi, but it can also be observed in the GPT series. This suggests that the visual nature of charts may interact with the language of the query in some ways, or that the training data for these MLLMs has a strong Chinese-language component.

**Performance using Chain-of-Thought strategy.** The application of a CoT strategy brings consistent performance gains for SOTA commercial models such as GPT-4o, Gemini-2.5-Pro, and Qwen-VL-MAX, improving results across most evaluation dimensions. The enhancement is particularly pronounced on tasks that demand precise interpretation of chart data and subsequent logical reasoning or calculation, including "Approximate Value", "Numerical", and "Judge" tasks. For example, the score for Gemini-2.5-Pro on the "Approximate Value" task doubled from 15.62% to 31.02%. This indicates that CoT effectively guides the model in deconstructing complex problems into manageable steps, thus increasing accuracy. However, the efficacy of CoT is not universal and is highly dependent on the model. A crucial finding is that, for many open-source models, employing a CoT strategy led to a significant performance degradation on the "Unanswerable" and "Open-Ended (sentence)" tasks. As a notable example, the accuracy of GLM-4.1V-9B in the "Unanswerable"

task plummeted from 35.71% to 17.27%. This reveals that CoT's effectiveness is deeply linked to a model's ability to suppress hallucinations and follow instructions. For models that lack specific fine-tuning on CoT-style data or possess insufficient reasoning abilities, forcing a step-by-step thought process can introduce interference, leading to logical confusion or an outright failure to produce a final answer. Furthermore, the impact of CoT varies between different types of tasks. It excels in tasks that require deep reasoning, but is less effective and even harmful for tasks with simple information extraction. For example, while GPT-4o's performance on "Approximate Value" improved by more than 12%, its score on "Open-Ended (vocabulary)" slightly decreased. This suggests that for simple, direct queries, the additional inferential steps introduced by CoT are unnecessary and may increase the risk of error highlighting the need for a dynamic prompting strategy in practical applications.

**Performance on Chart-Specific Models.** The results reveal that common MLLMs consistently outperform models specifically designed or fine-tuned for charts. This superiority is maintained across most tasks and persists regardless of whether CoT prompting strategies are used. While chart-specific models are highly optimized for existing benchmarks, the strong performance on curated datasets does not translate to the complex real-world document question-answering. Consequently, we think a more promising direction for future research is how to effectively adapt the powerful, generalizable abilities of foundation models to the document QA domain. The goal should be to leverage and enhance their core analytical capabilities for this task, rather than building specialized models that may lack real-world applicability.

**Key Insights and Observations.** Our experimental evaluation yields several critical insights into the current state of multi-chart question-answering. (1) Top-tier commercial models are the most capable and balanced performers. However, even these leading models struggle with numerical precision and complex reasoning, showing there is still significant room for improvement. (2) The open-source models present a wide spectrum of capabilities. While a few large models are competitive, the majority are not yet equipped to handle complex multi-chart reasoning tasks, indicating that further research and scaling are needed to close the performance gap. (3) Models explicitly trained for chart understanding did not outperform general-purpose MLLMs. This suggests that the ability to reason over complex visual data is more dependent on the scale of the foundational model and general reasoning capabilities than on narrow, task-specific training. (4) The most significant performance drops across all models occurred in tasks that required multi-step reasoning, numerical computation, and cross-chart comparisons. Future research should focus on enhancing these deep reasoning abilities to unlock the next level of performance in visual data understanding. (5) For complex chart analysis, CoT is a useful technique for achieving model's full potential. However, CoT prompts must be customized and optimized for specific models. Directly applying a prompt designed for a model like GPT-4 to an open-source alternative is likely to be counterproductive. (6) By further analyzing specific failure cases, we find that the models' failures are not merely due to visual perception issues, but more profoundly stem from a lack of cognitive capabilities such as working memory and multi-step planning. Many questions within ChartNexus require the model to perform multi-hop to compare data and to understand the implicit logic embedded within the context. This presents a significant challenge to the models' logical discrimination and reasoning abilities.

### 4.3 Diagnostic Analysis: Boundaries and Bottlenecks

To rigorously pinpoint the limitations of current MLLMs within the ChartNexus, we conducted a series of fine-grained ablation studies and stratified analyses. These diagnostics reveal three fundamental bottlenecks: spatial projection failures, semantic grounding dependency, and resolution constraints in dense visual contexts.

**The Dimensionality Barrier in Visual Encoders.** Our stratified performance analysis by chart type (see Appendix G.1) exposes a critical deficiency in handling spatial information. While SOTA models demonstrate robustness on 2D charts, we observe a significant "performance drop" when processing 3D charts. As detailed in Table 8, Even the best-performing model (Qwen2.5-VL-32B) falling to 36.1%. This universal degradation suggests that current vision encoders, predominantly pre-trained on 2D web images, struggle to re-

solve the projection loss inherent in rendering 3D data onto a 2D plane. The models fail to accurately perceive depth and perspective, leading to severe hallucinations in reading data points from 3D axes.

**The Necessity of Cross-Modal Semantic Bridging.** ChartNexus is designed to simulate real-world document analysis where charts rarely exist alone. To quantify the models' reliance on textual context, we performed an ablation study removing all captions and surrounding paragraphs (see Appendix G.2). The results reveal a sharp "Context Gap," with performance degrading in the "No-Context" setting. For instance, Qwen2.5-VL-7B's accuracy on Open-Ended Vocabulary tasks plummeted from 46.67% to 15.1%. This confirms that MLLMs do not merely "see" the chart; they rely heavily on textual cues to disambiguate visual features and ground their reasoning. The text serves as a semantic bridge; without it, the models struggle to infer the implicit logic and domain-specific nuances required for complex reasoning.

**Visual Resolution vs. Reasoning Capacity.** We further disentangled whether errors in multi-chart tasks stem from logical complexity (reasoning across entities) or visual density (perception limits). By comparing performance on composite images (single image containing multiple subplots) versus multiple discrete image files (see Appendix G.3), we identified a distinct scaling law. Smaller models, such as InternVL3-14B, suffer a massive performance drop of 23.5% when processing composite subplots compared to discrete images. This highlights a "resolution bottleneck": when multiple charts are packed into a single token sequence or image patch grid, the effective resolution per chart diminishes, overwhelming the encoder's capacity. In contrast, larger models (e.g., InternVL3-38B) maintain robustness across both settings, suggesting that increased parameter scale correlates with a superior ability to attend to fine-grained visual details within dense information streams.

**Hallucination and Format Bias.** We further investigate model faithfulness by analyzing "Unanswerable" questions, defining a hallucination as providing a specific answer when the ground truth is "Unanswerable". Our analysis reveals a severe "Selection Bias" in the Multiple-Choice (MC) format compared to the Judgment (Boolean) format. As detailed in Appendix G.4, models like GLM-4.1V-9B and Qwen2.5-VL-32B achieved 0 successful refusals ($TP = 0$) in the MC setting, hallucinating an answer in 100% of unanswerable cases, whereas they demonstrated some capacity to refuse in the Judgment format correctly. This dissociation—recognizing a lack of evidence in one format while generating an answer in another—suggests that hallucinations are not mainly due to visual encoding failures. Instead, they likely stem from structural biases in the pre-training corpus, which lead models to follow selection formats instead of rigorously verifying premises.

## 5 Conclusion

This study introduces ChartNexus, a novel and challenging multi-chart question-answering benchmark that addresses a critical gap in evaluating MLLMs for real-world document analysis scenarios. Unlike existing benchmarks that focus on isolated chart understanding, ChartNexus evaluates models' ability to synthesize information across multiple interrelated charts within authentic document contexts, incorporating surrounding textual information and complex reasoning chains. Our benchmark comprises 6,793 real-world charts and 1,370 meticulously human-annotated question-answer pairs, systematically organized through a comprehensive taxonomy. Our evaluation of 23 state-of-the-art MLLMs reveals substantial limitations in current multi-chart reasoning capabilities. While leading models achieve over 90% accuracy on single-chart benchmarks, their performance drops by more than half on ChartNexus, demonstrating that multi-chart reasoning remains a largely unsolved challenge. Through systematic failure analysis, we identify critical weaknesses in models' ability to retain information across multiple visualizations, perform cross-modal reasoning, and execute multi-hop inferences. By shifting evaluation focus from isolated visual perception to complex cross-modal synthesis, ChartNexus provides essential diagnostic tools for advancing MLLM development and serves as a roadmap for developing more capable models for authentic document analysis scenarios.

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

APPENDIX

## A    DATA ANNOTATION

### A.1    DATA COLLECTION PRINCIPLES

The construction of our benchmark was predicated on a set of rigorous principles designed to ensure its validity, relevance, and robustness for evaluating the chart-to-code generation capabilities of MLLMs.

**Mitigation of Data Leakage through Novel Data Sourcing.** A primary consideration was the reduction of potential data leakage, wherein a model's performance could be artificially inflated due to the inclusion of benchmark data in its pre-training corpus. To counteract this, we deliberately avoided common online repositories and auto-generated examples. Instead, our dataset was exclusively curated from contemporary and domain-specific sources, including academic papers from arXiv, economic reports from the World Bank[1] and the Organisation for Economic Co-operation and Development (OECD)[2], sociological studies from the Pew Research Center[3], Statista[4], various public government datasets[5][6] and industries research reports, including the China Internet Network Information Center (CNNIC)[7], Communications World[8], and the National Consortium for the Study of Terrorism and Responses to Terrorism (START)[9]. This methodology ensures that the benchmark serves as a true test of a model's generalization and reasoning abilities.

**Adherence to Real-World Application Scenarios.** The benchmark is designed to reflect the authentic data visualization requirements of users in practical settings. By sourcing charts directly from academic, financial, and governmental publications, we ensure that each task is grounded in a genuine use case. This alignment with real-world scenarios enables a more precise and relevant evaluation of LMMs, steering their development toward greater utility in professional and research contexts.

**Comprehensive Coverage of Chart Type and Topic.** Our sourcing strategy naturally produces a dataset with significant diversity in both chart typology and complexity. The collection intentionally moves beyond rudimentary chart types (e.g., simple bar, line, and pie charts) to encompass a wide spectrum of visualizations used in specialized fields. Furthermore, the benchmark includes charts with varying levels of information density and structural complexity, from single-series plots to multi-faceted figures with composite elements. This ensures a thorough assessment of a model's ability to handle a wide range of visualization challenges.

### A.2    DATA ANNOTATION PRINCIPLES AND PIPELINE

#### A.2.1    PRINCIPLES

**Emulation of Authentic User Inquiries.** All questions must be framed to reflect plausible, real-world scenarios. The objective is to simulate the analytical tasks a user would perform when encountering a multi-chart figure. Therefore, questions are designed to be pragmatic, focusing on core analytical goals such as comparison, trend identification, summarization, or anomaly detection. Abstract or contrived questions that do not correspond to a genuine analytical intent are explicitly disallowed.

**Mandatory Synthesis of Multi-Chart Information.** A fundamental criterion is that every question must necessitate the integration of information from two or more individual charts to be answered correctly. Questions that can be resolved by analyzing a single sub-

---

[1]https://openknowledge.worldbank.org
[2]https://www.oecd.org/en.html
[3]https://www.pewresearch.org/publications
[4]https://www.statista.com
[5]https://www.stats.gov.cn/sj/zxfb
[6]https://www.guizhou.gov.cn/zwgk/zfsj/tjgb
[7]https://www.cnnic.cn/6/180/index.html
[8]https://www.cww.net.cn/subjects/cha/download
[9]https://www.start.umd.edu/publications

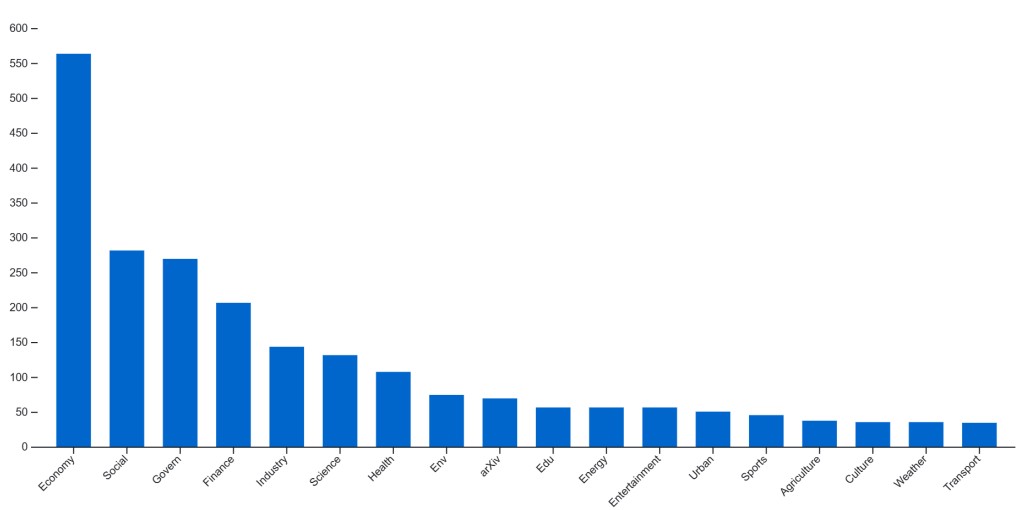

Figure 2: Different chart types in ChartNexus.

chart in isolation are considered invalid for this benchmark. This principle ensures that the tasks specifically target the model's capability for cross-referencing and synthesizing data from disparate visual sources within a single scene. For example, a valid question might ask to correlate the trend in a line chart with the composition shown in a corresponding pie chart.

**Requirement for Contextual Understanding in Complex Reasoning.** For questions categorized as requiring complex reasoning, the model must do more than simply extract and compare data points. These questions are constructed to require the integration of contextual information derived from the figure's title, caption, or other textual elements. The answer should depend on a holistic understanding of the scene, compelling the model to, for instance, explain a trend visible in the charts by referencing a cause mentioned in the accompanying text. This tests a deeper level of multimodal comprehension beyond basic visual data retrieval.

### A.2.2 Pipeline

**Automated Data Pre-processing Pipeline.** The initial stage involved the automated extraction and structuring of chart-centric data from raw PDF documents. First, each source document was parsed into Markdown format using the Mineru library. Following this, a crucial filtering step was executed where the Qwen2.5-VL model programmatically analyzed all extracted images, identifying and discarding those irrelevant to the ChartNexus theme, such as natural photographs or schematic diagrams. The refined Markdown content was then reconstituted into a structured JSON format using markdown-it-py. In the final pre-processing step, a hybrid approach was utilized combining rule-based heuristics and the Qwen3 model to extract salient contextual information (e.g. captions and surrounding paragraphs) associated with each chart. This automated pipeline resulted in a high-quality candidate dataset primed for human annotation.

**Pilot Annotation:** We use Label Studio[10] to construct the annotation tasks, allowing for iterative refinement of the requirements. Initially, graduate students with backgrounds in data analysis and deep learning conduct a pilot annotation. Through this process, we finalize the necessary annotation items for the benchmark. Based on this experience, we categorize the QA formats into five types: multiple-choice, judgment, vocabulary-answer,

---

[10]A labeling platform: `https://labelstud.io`

numerical estimation, and open-ended questions. In addition to the QA pair, annotators were required to specify the question's difficulty level and its key difficulties. The pipeline of our annotations is illustrated in the corresponding Figure 1.

**Reference Question Generation:** In the formal annotation phase, we summarize the question templates from the pilot stage. These manually crafted seed questions served as few-shot examples for an LLM. The model was instructed to mimic the reasoning patterns of these examples and generate multiple sets of candidate questions for each task based on the provided charts and context, offering a convenient starting point for human annotators.

**Manual Question Annotation:** We recruit well-trained annotators and provide them with a meticulous annotation guide. They are tasked with either refining the questions generated by an LLM based on specific chart information or using these reference templates as inspiration to formulate new questions with greater reasoning depth. This process ensures that each question is closely related to the charts, logically self-consistent, and requires the synthesis of information from at least two charts.

**Answer Annotation:** Subsequently, annotators are required to answer these questions and write the corresponding ground-truth answers. It is important to note that not all annotated QA pairs are answerable; a portion of the questions is intentionally designed to be unanswerable based on the provided charts.

**Detailed Annotation Schema.** The annotation process was systematically divided into two primary, sequential tasks: chart-level annotation and QA pair annotation.

A. Chart Annotation Task: This initial task focused on the structural and typological properties of the visual elements. Annotators were required to label the primary chart type (e.g., bar, line, scatter plot) and determine if the image contained sub-charts, quantifying them if present.

B. Question-Answer Pair Annotation Task: This second, more complex task involved assessing and labeling the generated QA pairs. Annotators were required to provide multiple labels for each pair:

Suitability: A binary judgment on whether the associated chart combination is appropriate for formulating a reasonable and unambiguous question.

Answer Type: Classification of the correct answer's format, categorized as Numerical, Open-Ended, Boolean (True/False), or Multiple Choice.

Reasoning Skill: Identification of the core challenge or difficulty element the question targets, such as Numerical Calculation, Visual Grounding (locating specific elements), or Comparative Reasoning (comparing trends across charts).

Answerability and Difficulty: A final assessment indicating if the question is answerable given the provided context, accompanied by a quantitative score representing its overall difficulty.

## B  Chart Examples

This section presents each chart type in ChartNexus. ChartNexus comprises a total of 3,198 distinct image files sourced from real-world documents. However, to accurately reflect the information density models must process, we define our atomic unit as a "semantic chart." Since approximately 16.69% of our images are composite figures containing multiple subplots (averaging 4.78 subplots per composite image), the total count of atomic semantic charts is 6,793. This distinction is crucial, as reasoning often requires extracting specific data from a single subplot within a dense composite figure.

ChartNexus encompasses a structure of 17 types. The categories comprise of: Line, Bar, Pie, Scatter, Radar, Candlestick, Boxplot, Heatmap, Graph, Tree, Sunburst, Sankey, Funnel, 3D, Area, and Table. Here are some examples for different chart types in our ChartNexus.

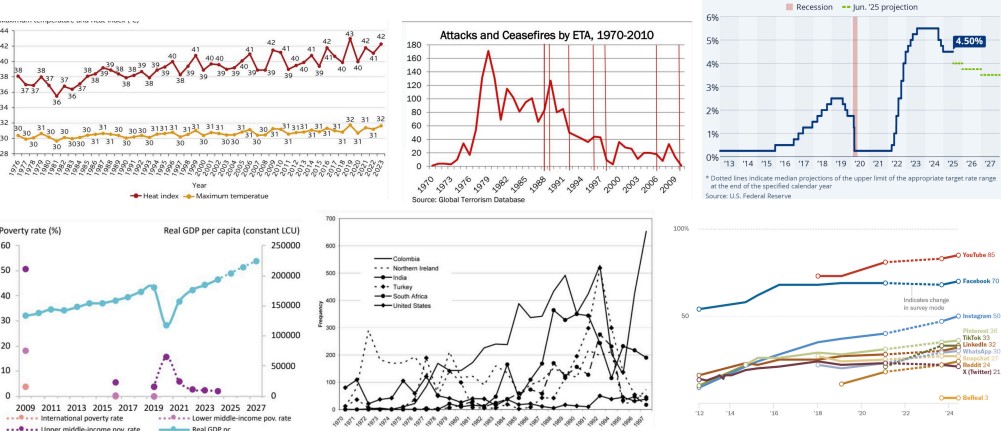

Figure 3: Examples of Line Charts in ChartNexus

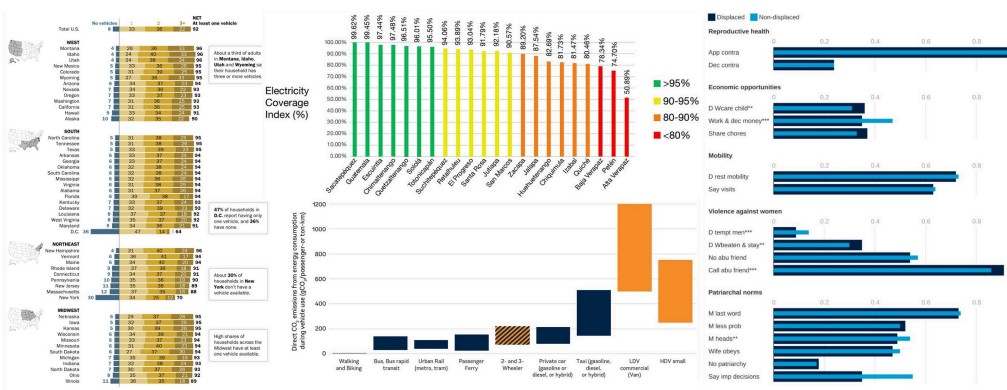

Figure 4: Examples of Bar Charts in ChartNexus

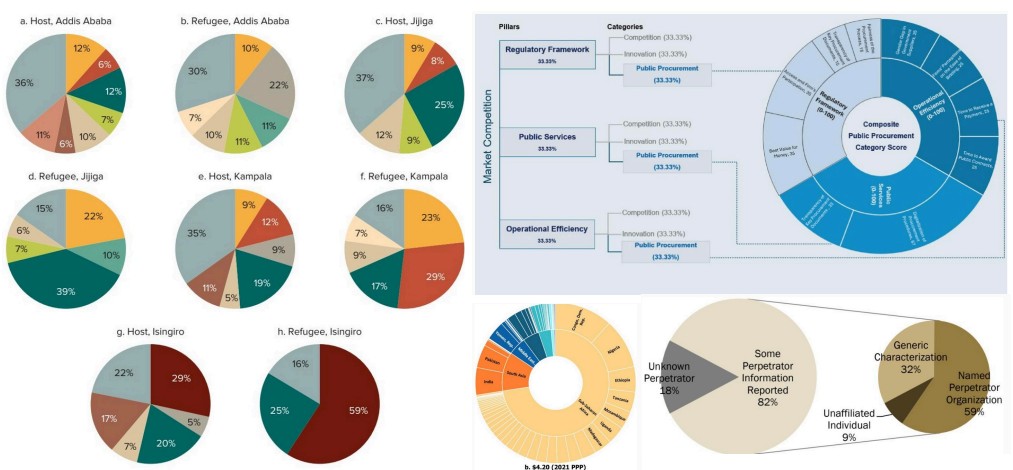

Figure 5: Examples of Pie Charts in ChartNexus

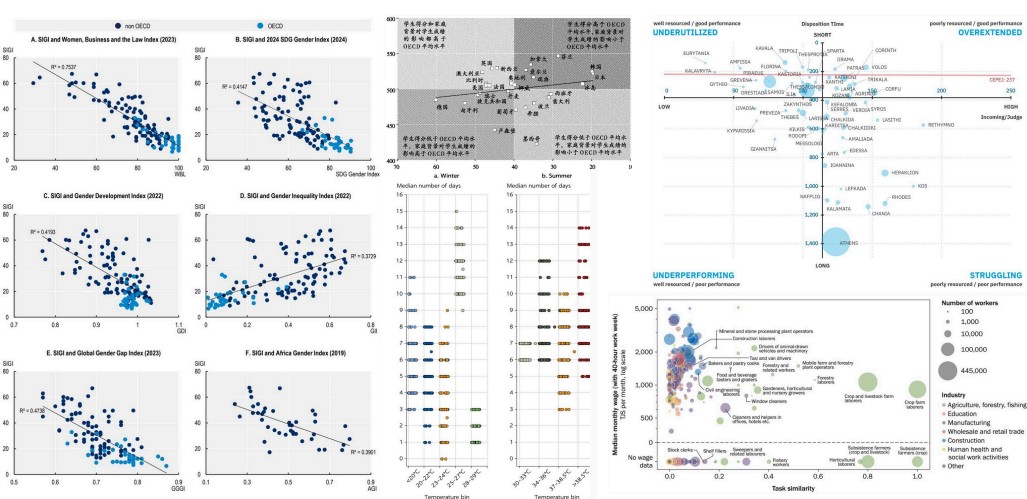

Figure 6: Examples of Scatter Charts in ChartNexus

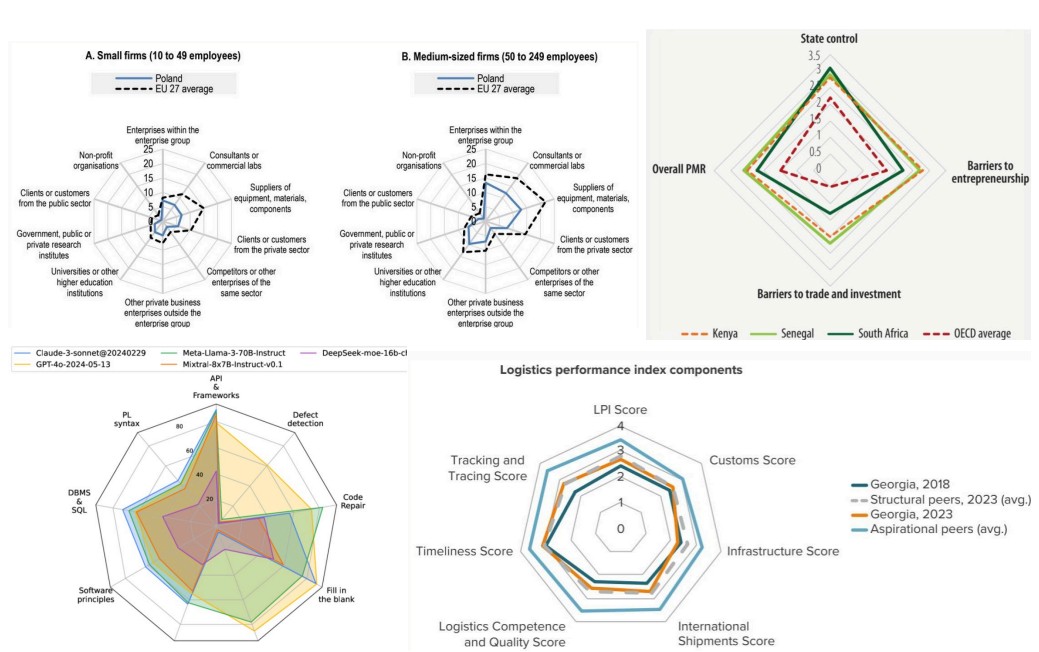

Figure 7: Examples of Radar Charts in ChartNexus

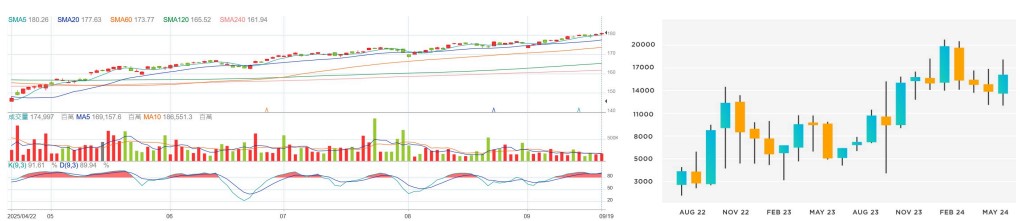

Figure 8: Examples of Candlestick Charts in ChartNexus

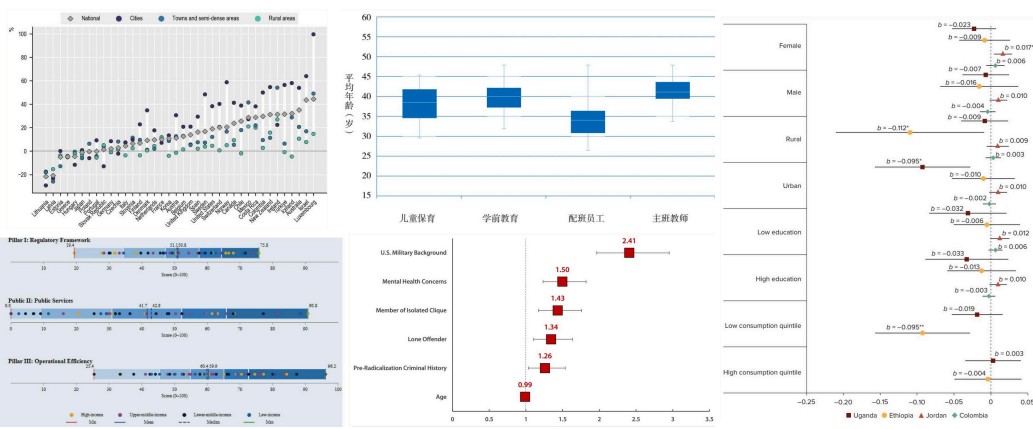

Figure 9: Examples of Boxplot Charts in ChartNexus

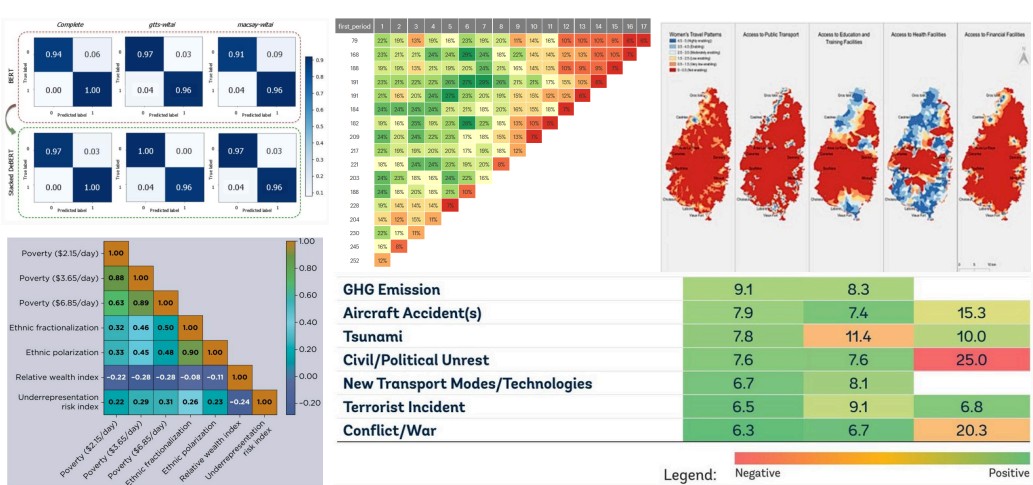

Figure 10: Examples of Heatmap Charts in ChartNexus

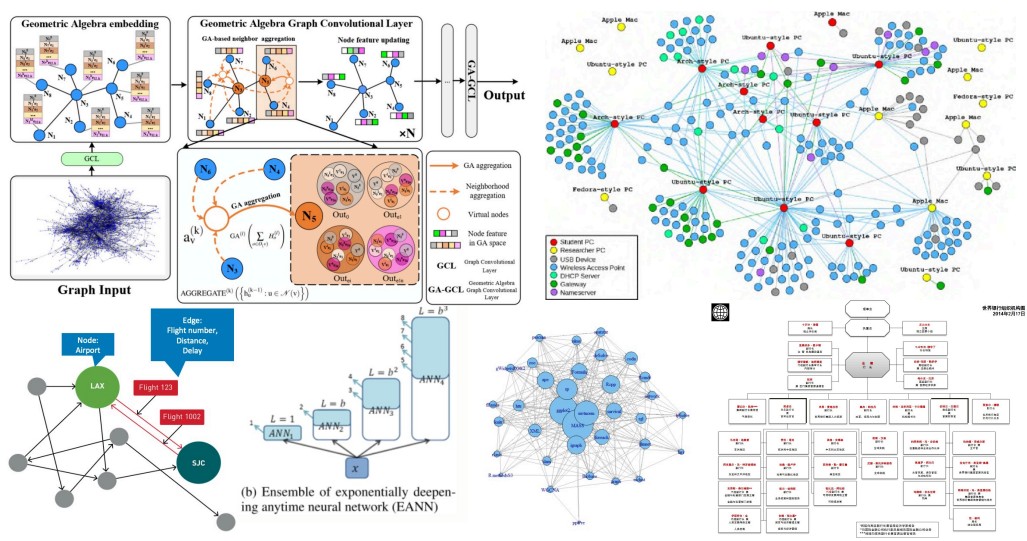

Figure 11: Examples of Graph Charts in ChartNexus

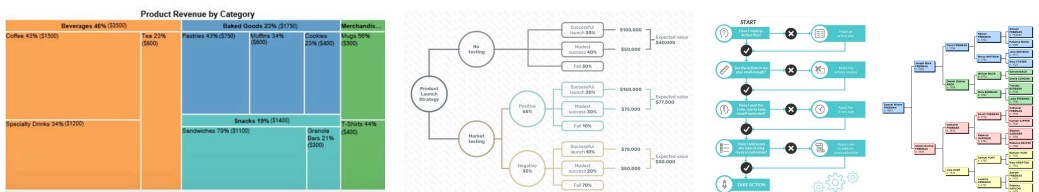

Figure 12: Examples of Tree and Treemap Charts in ChartNexus

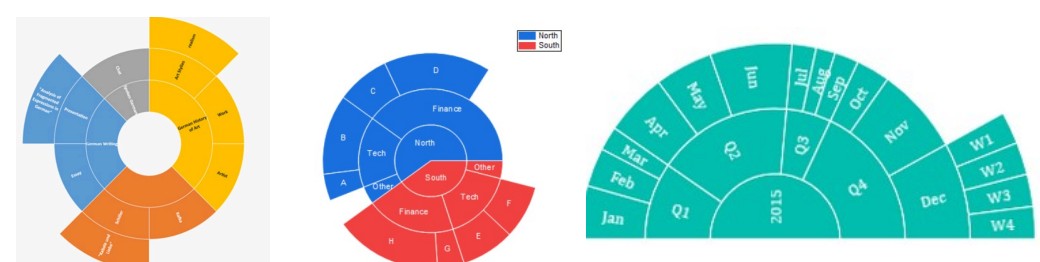

Figure 13: Examples of Sunburst Charts in ChartNexus

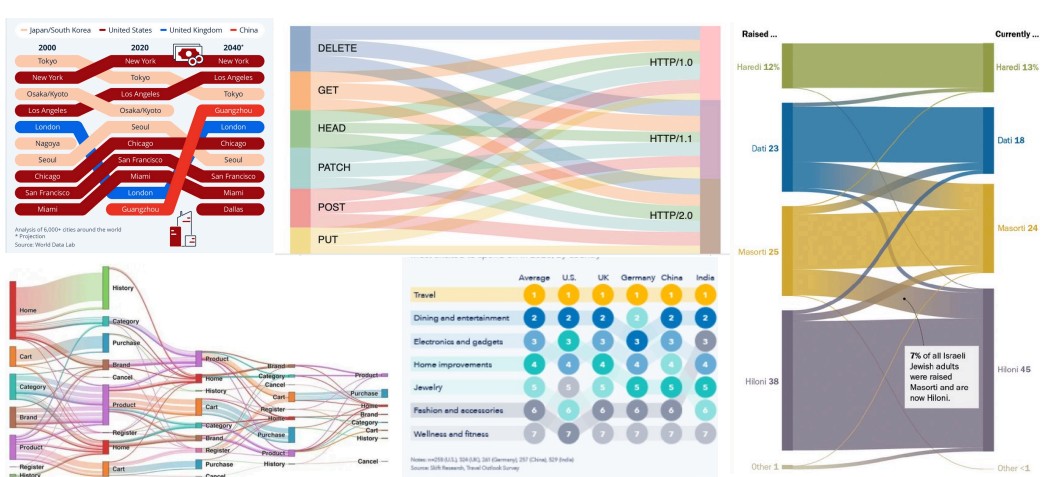

Figure 14: Examples of Sankey Charts in ChartNexus

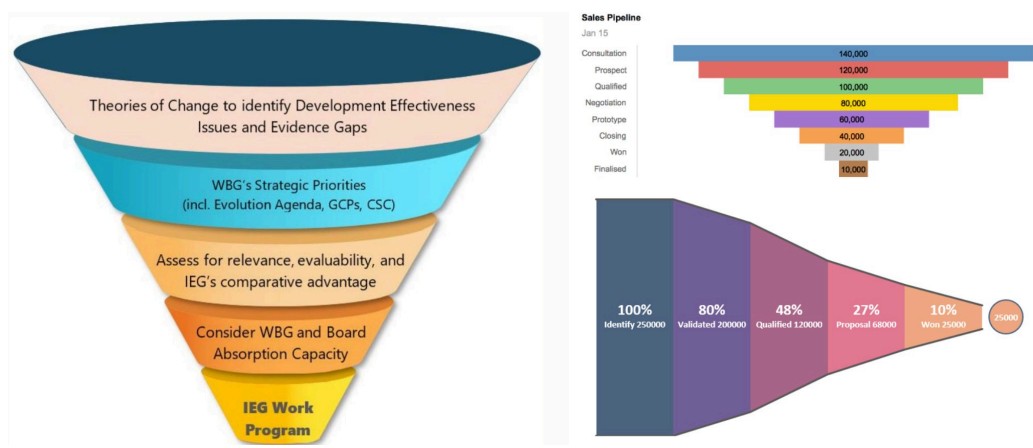

Figure 15: Examples of Funnel Charts in ChartNexus

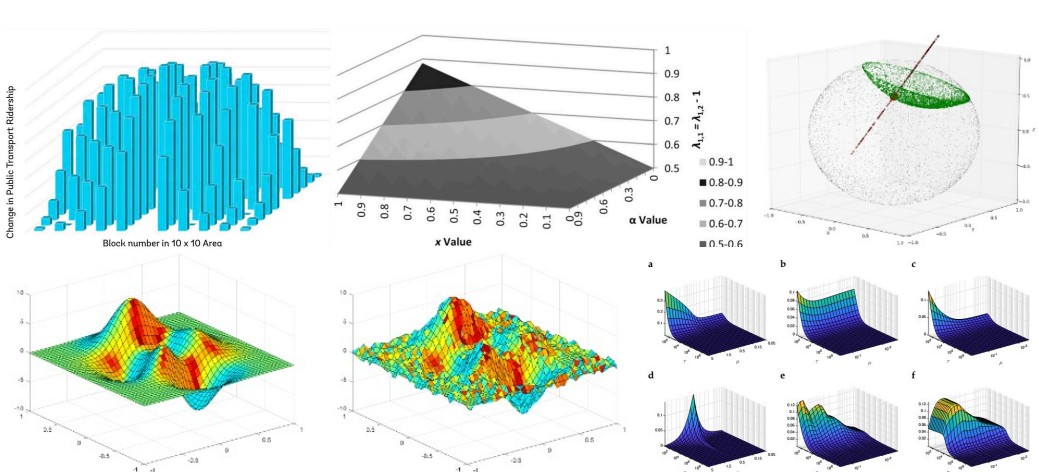

Figure 16: Examples of 3D Charts in ChartNexus

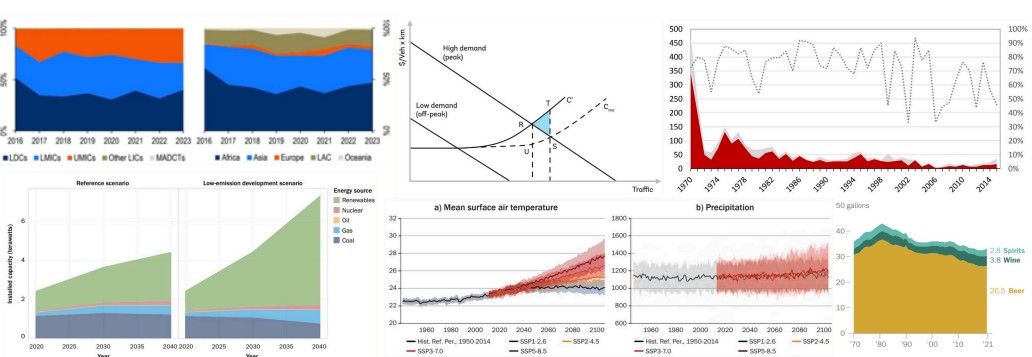

Figure 17: Examples of Area Charts in ChartNexus

Figure 18: Examples of Tables in ChartNexus

## C   DETAILS OF EVALUATION

In this section, we present the details of evaluation models and metrics, including prompts in calculating Accuracy and F1 Score from SEAT Method (Zhu et al., 2025b).

**Models.** For commercial models, we utilize the official APIs to access their stable versions. For open-source models, the model weights can be downloaded from the links provided below:

Table 7: List of Open-Source Models and Download Links

| Model Name | URL |
|---|---|
| SmolVLM-2.3B | https://huggingface.co/HuggingFaceTB/SmolVLM-Instruct |
| Phi-4-multimodal-Instruct | https://huggingface.co/microsoft/Phi-4-multimodal-instruct |
| Bagel | https://huggingface.co/ByteDance-Seed/BAGEL-7B-MoT |
| Kimi-VL-A3B-Thinking | https://huggingface.co/moonshotai/Kimi-VL-A3B-Thinking |
| Qwen2.5-VL-7B | https://huggingface.co/Qwen/Qwen2.5-VL-7B-Instruct |
| GLM-4.1V-9B | https://huggingface.co/zai-org/GLM-4.1V-9B-Thinking |
| InternVL3-14B | https://huggingface.co/OpenGVLab/InternVL3-14B |
| Qwen2.5-VL-32B | https://huggingface.co/Qwen/Qwen2.5-VL-32B-Instruct |
| InternVL3-38B | https://huggingface.co/OpenGVLab/InternVL3-38B |
| ChartGemma | https://huggingface.co/ahmed-masry/chartgemma |
| ChartInstruct-LLama2 | https://huggingface.co/ahmed-masry/ChartInstruct-LLama2 |
| ChartMoe | https://huggingface.co/IDEA-FinAI/chartmoe |

**Accuracy for Close-Ended Questions.** For question categories with definitive, single-ground-truth answers—specifically Multiple Choice, True/False, Numerical Calculation, and Open-Ended(vocabulary) (single word/phrase) questions—we utilize Accuracy as the primary evaluation metric. A model's response is considered correct only if it exactly matches the ground-truth answer. The overall accuracy is calculated as:

$$\text{Accuracy} = \frac{\text{Number of Correct Predictions}}{\text{Total Number of Questions}} \tag{1}$$

This strict metric is appropriate for tasks where precision is unambiguous and semantic variance is not a factor.

**F1 Score for Open-Ended Questions.** For open-ended questions that require a full sentence as an answer, a direct string match is often inadequate, as semantically equivalent responses can have different phrasings. To account for this, we evaluate these responses using the **F1 Score**, which provides a balanced measure of precision and recall. The calculation is facilitated by the **SEAT** methodology, which is designed to handle semantic similarities.

To standardize the evaluation, we first process the model's raw generation to isolate the final answer. This is achieved using a specifically designed extraction prompt, presented below:

The following are prompts for evaluating the model's output.

**General prompt for extracting predict answer from model's response**

You will be given a question about some charts. You need to answer this question based on the provided charts as well as its related context. The context corresponding to each chart will be placed within <context> </context> tags, and the question to be answered will be placed within <question></question> tags.

Your answer should be a single word, number, or phrase. If the question is unanswerable based on the information in the provided image, your answer should be unanswerable.

Do not generate units. But if numerical units such as million, m, billion, B, or K are required, use the exact notation shown in the chart. If there are multiple answers, put them in brackets using this format ["Answer1", "Answer2"].

Figure 19: General prompt for extracting the answer from the model's output, which will pass to Qwen3-32B for extraction.

**General prompt for evaluating answer**

**System Prompt:**

You are a helpful assistant. You need to compare a given answer with the ground truth to determine if it is correct. Always place your final answer within <answer></answer> tags.

**User Prompt:**

You are required to determine if a predicted answer is correct when compared with the ground truth. The question will be placed within <question></question> tags, predicted answer will be placed within <predict> </predict> tags, and the ground truth answer will be placed within <gt></gt> tags.

The predicted answer may contain some thought or reasoning content in addition to the final answer. You must first find the correct answer: a word, phrase, or number within the prediction, and then compare it with the ground truth.

Remember to only respond with 'true' or 'false', and place your judgment within <answer></answer> tags.

Question: <question>{question}</question>

Predict Answer: <predict>{predict}</predict>

Ground Truth: <gt>{gt}</gt>

Figure 20: Prompt to evaluate model's response.

**Evaluating Multi-Choice questions**

You are required to determine if a predicted answer is correct when compared with the ground truth. The question will be placed within <question></question> tags, predicted answer will be placed within <predict></predict> tags, and the ground truth answer will be placed within <gt></gt> tags.

The predicted answer may contain additional content, such as reasoning, besides the final answer. You must first extract the correct answer from within the prediction. The answer should be a single multiple-choice option (e.g., A, B, C, etc.). You should then compare this extracted option with the ground truth.

Remember to only respond with 'true' or 'false', and place your judgment within <answer></answer> tags.

Figure 21: Prompt to evaluate Multi-Choice questions.

**Evaluating Numerical-Calculation questions**

You are required to determine if a predicted answer is correct when compared with the ground truth. The question will be placed within <question></question> tags, predicted answer will be placed within <predict></predict> tags, and the ground truth answer will be placed within <gt></gt> tags.

The predicted answer may contain additional content, such as reasoning, besides the final answer. You must first extract the correct answer from within the prediction, which should be an estimated numerical value. You should then compare this extracted number with the ground truth.

The predicted numerical value is considered correct if it is within a 5% margin of error relative to the ground truth value.

Remember to only respond with 'true' or 'false', and place your judgment within <answer></answer> tags.

Figure 22: Prompt to evaluate Numerical questions.

**Evaluating True/False questions**

You are required to determine if a predicted answer is correct when compared with the ground truth. The question will be placed within <question></question> tags, predicted answer will be placed within <predict></predict> tags, and the ground truth answer will be placed within <gt></gt> tags.

The predicted answer may contain additional content, such as reasoning, besides the final answer. You must first extract the correct answer from within the prediction. The answer should be a response to a true/false or yes/no type of question (e.g., 'true', 'false', 'yes', 'no').

Remember to only respond with 'true' or 'false', and place your judgment within <answer></answer> tags.

Figure 23: Prompt to evaluate True/False questions.

**Evaluating Open-End questions by SEAT**

## 目标

请将大模型的回答与用户提供的参考答案进行对比，步骤如下：

1. 提取关键答案

  1. 定位大模型回答的"最终总结"，逐个对照参考答案中的子问题，从大模型的"最终总结"中提取每个问题对应所有关键回答。关键回答应仅包含核心的、直接回答问题的内容。

  2. 对已识别出的某个关键回答进行补充说明的内容，应与该关键回答合并为一个整体，不要拆分成新的答案要素。只有在内容明显独立、可与参考答案中不同要素相对应时，才视为新答案。

2. 对比并标注：将提取出的回答与参考答案逐一对比，按以下标准进行标注：

  1. 错误答案（false）：如果大模型多输出了一些要素，并且这些要素与参考答案无法对应或仅是多余的补充信息（不是在同一个要素中补充，而是产生了多余答案要素），则判定为错误。

  2. 正确答案（true）：如果该条回答与参考答案某一要素含义一致或高度吻合，视为正确。

  3. 注意：每个从大模型回答中提取出的答案要素，都要有相应的 true 或 false 标签，确保每个回答要素都被检查。

### 输出格式

```\n{{\n "问题列表": [\n {{\n  "问题": "子问题1",\n  "参考答案": ["答案1", "答案2"],\n  "大模型的回答": ["关键回答1", "关键回答2"],\n  "是否正确": [true, false]\n }},\n {{\n  "问题": "子问题2",\n  "参考答案": ["答案1"],\n  "大模型的回答": ["关键回答1", "关键回答2", "关键回答3", "关键回答4"],\n  "是否正确": [false, true, false, false]\n }}\n ]\n}}\n```

### 参考答案

{answer}

Figure 24: Prompt to evaluate Open-Ended(sentence) questions by SEAT method.

## D    Model Configurations and Prompting Methods

### D.1    Generation Configurations

For open-weight models, we set the temperature $\tau = 0.1$ to achieve optimal results, while for proprietary models, we set the temperature $\tau = 0$ for greedy decoding. For all models, we set the maximum generation length to 4096. Additionally, we use BF16 for model inference for open-weight models. All models are inferred on RTX 6000 Ada.

### D.2    Prompts

To investigate the model's reasoning capabilities, we conducted experiments using a Chain-of-Thought (CoT) prompting strategy. This approach was implemented by modifying the model's default system prompt to explicitly elicit a step-by-step reasoning process before providing a final answer.

The specific system prompt employed for our CoT experiments is detailed below:

---

**CoT prompt**

**System Prompt:**

You are a helpful assistant for a question-answering task.

Your goal is to answer the question based on the provided contexts.

First, think step-by-step and write down your reasoning process within <reasoning></reasoning> tags. This process should break down how you use the contexts to arrive at the answer.

Finally, provide your final answer within <answer></answer> tags.

---

Figure 25: Prompt for CoT experiments.

After the system prompt, the model is instructed to generate associated answer for the given question.

---

**Extracting predict answer for Multi-Choice questions**

You will be given a multiple-choice question about charts. You need to answer this question based on the provided charts and its related context. The context for each chart will be placed within <context></context> tags, and the question will be placed within <question></question> tags.

Your answer should be the letter of the correct option (e.g., A, B, C, etc.). If the question is unanswerable based on the information in the provided image, your answer should be unanswerable.

---

Figure 26: Prompt to extract answers from model's responses.

Then, the context, images and question will be fed into the model.

**Extracting predict answer for Numerical-Calculation questions**

You will be given a numerical question-answering task about charts. You are required to answer this question based on the provided charts and its related context. The context for each chart will be placed within <context></context> tags, and the question will be placed within <question></question> tags.

Your answer should be the most appropriate approximate numerical value. If the question is unanswerable based on the information in the provided image, your answer should be unanswerable.

Do not generate units. But if numerical units such as million, m, billion, B, or K are required, use the exact notation shown in the chart. If there are multiple answers, put them in brackets using this format ["Answer1", "Answer2"].

Figure 27: Prompt to extract answers for Numerical questions.

**Extracting predict answer for True/False questions**

You will be given a true/false question about charts. You are required to answer this question based on the provided charts and its related context. The context for each chart will be placed within <context></context> tags, and the question will be placed within <question></question> tags.

Your answer should be either 'true' or 'false'. If the question is unanswerable based on the information in the provided image, your answer should be unanswerable.

Figure 28: Prompt to extract answers for True/False questions.

**Extracting predict answer for Open-End questions**

You will be given an open-ended question about charts. You are required to answer this question based on the provided charts and its related context. The context for each chart will be placed within <context></context> tags, and the question will be placed within <question></question> tags.

Your answer should be a logical and well-reasoned explanation that addresses the question. If the question is unanswerable based on the information in the provided image, your answer should be unanswerable.

Figure 29: Prompt to extract answers for Open-Ended(vocabulary) questions.

# E  QA Examples

In this section, we use several examples in ChartNexus to illustrate our annotation methodology for different answer types and difficulty factors.

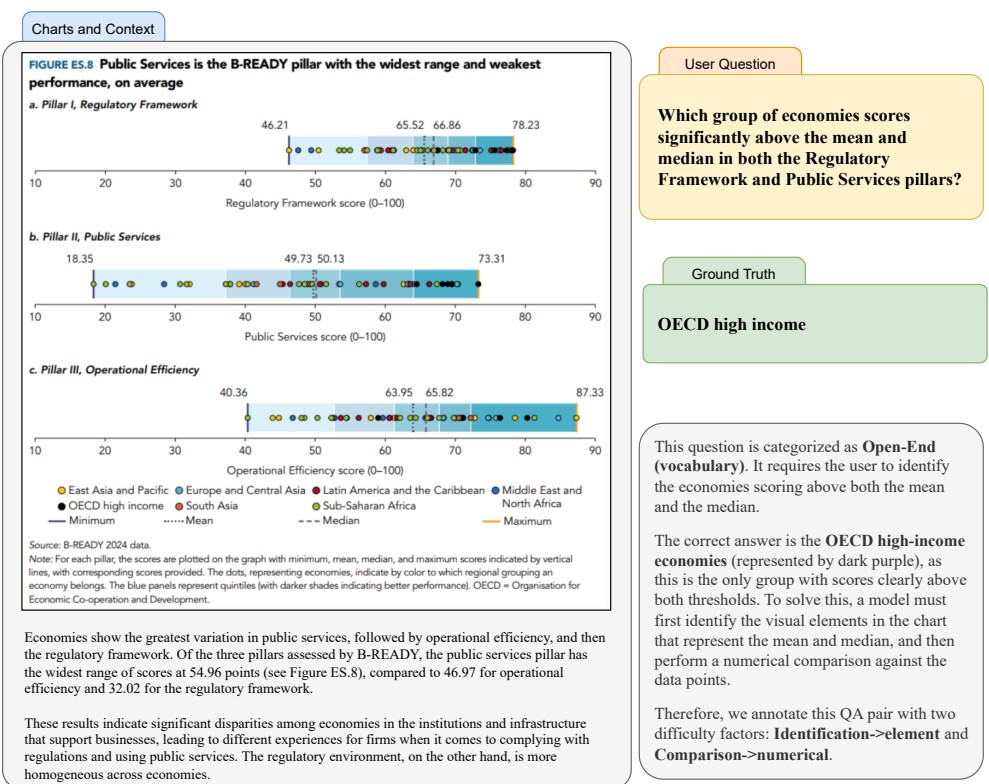

Figure 30: Example 1 of Annotations.

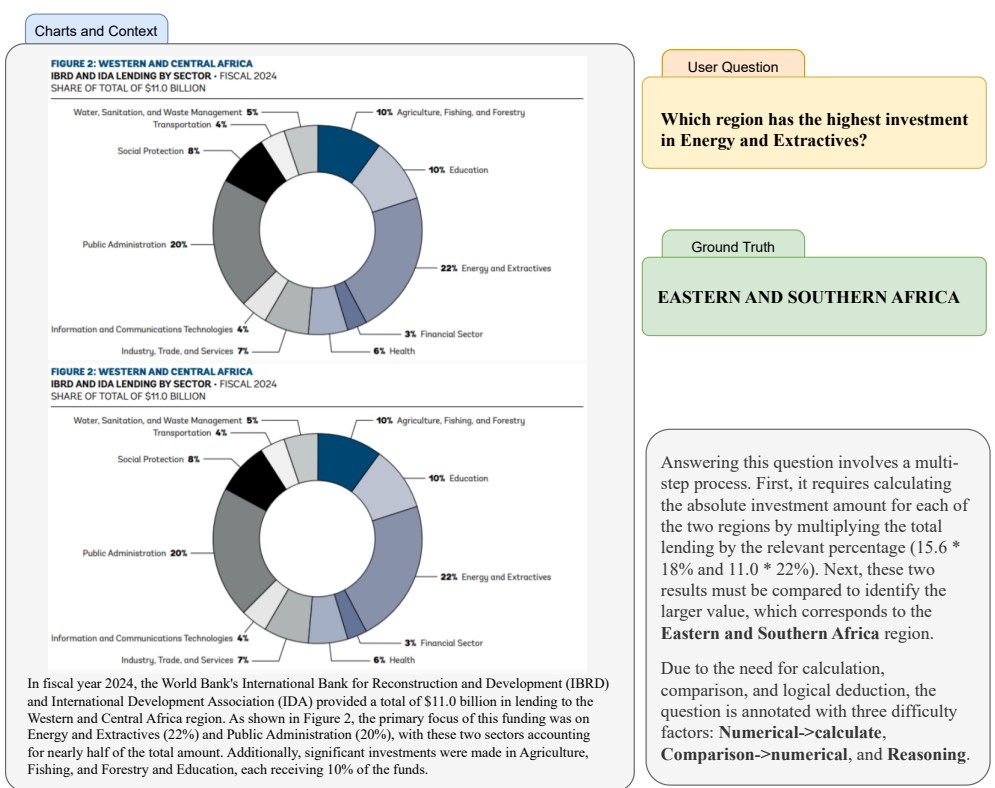

Figure 31: Example 2 of Annotations.

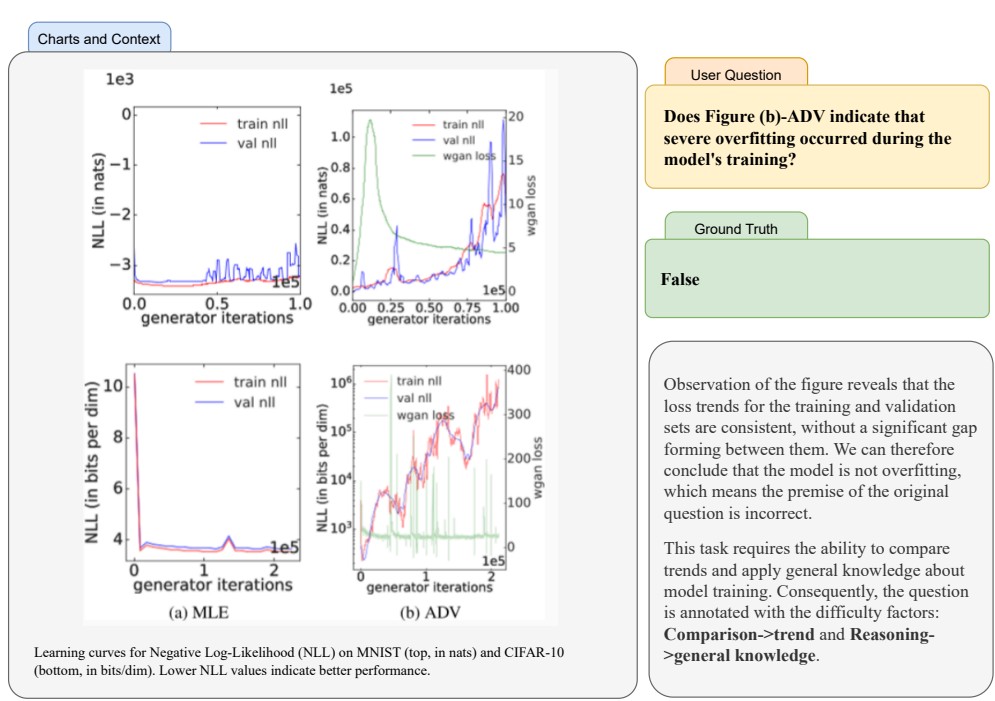

Figure 32: Example 3 of Annotations.

# F  Cases of Error Analysis

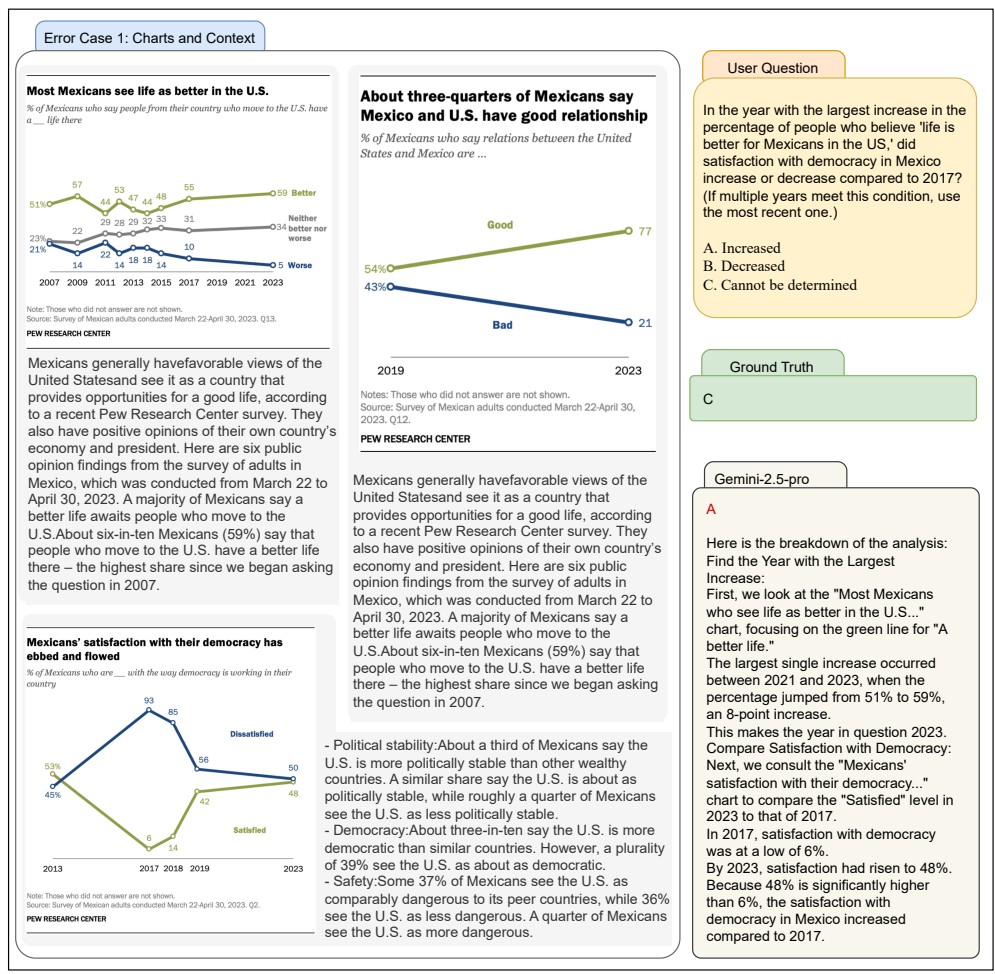

Figure 33: Error case 1. For this case, the correct procedure is to first identify the year with the largest increase for the "Better" metric in the top-left chart (which is approximately 2012). Then, this year's "satisfaction with democracy in Mexico" should be compared with the 2017 value. However, since the satisfaction data for 2012 is not provided, the correct answer is "C. Cannot be determined". Gemini-2.5-pro erred by using incorrect data from 2023 for comparison.

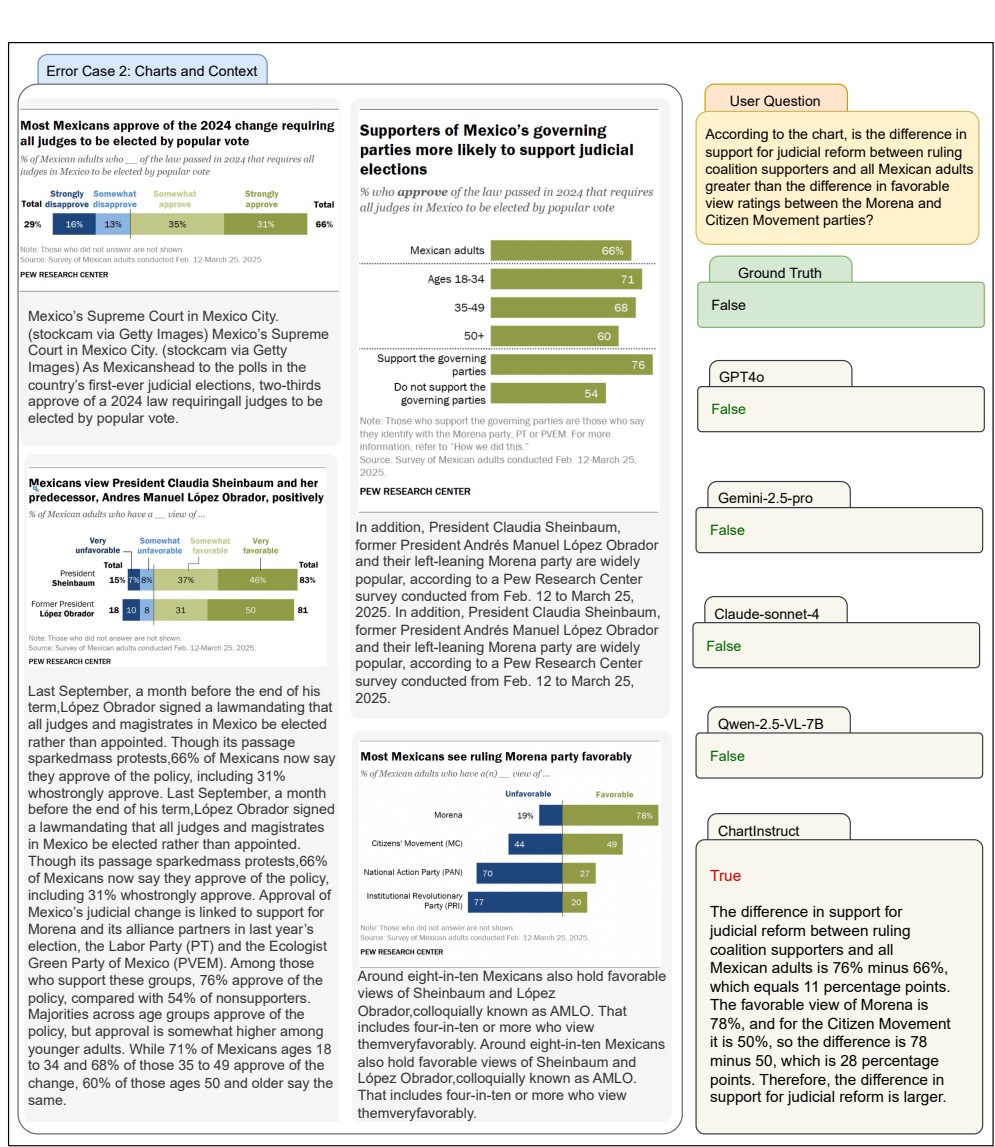

Figure 34: Error case 2. In this case, the general-purpose MLLMs successfully identified the required elements, whereas the specialized chart model, ChartInstruct, failed. This outcome suggests that models trained primarily on existing chart benchmarks may not be well-suited for the complexities of real-world document question-answering scenarios.

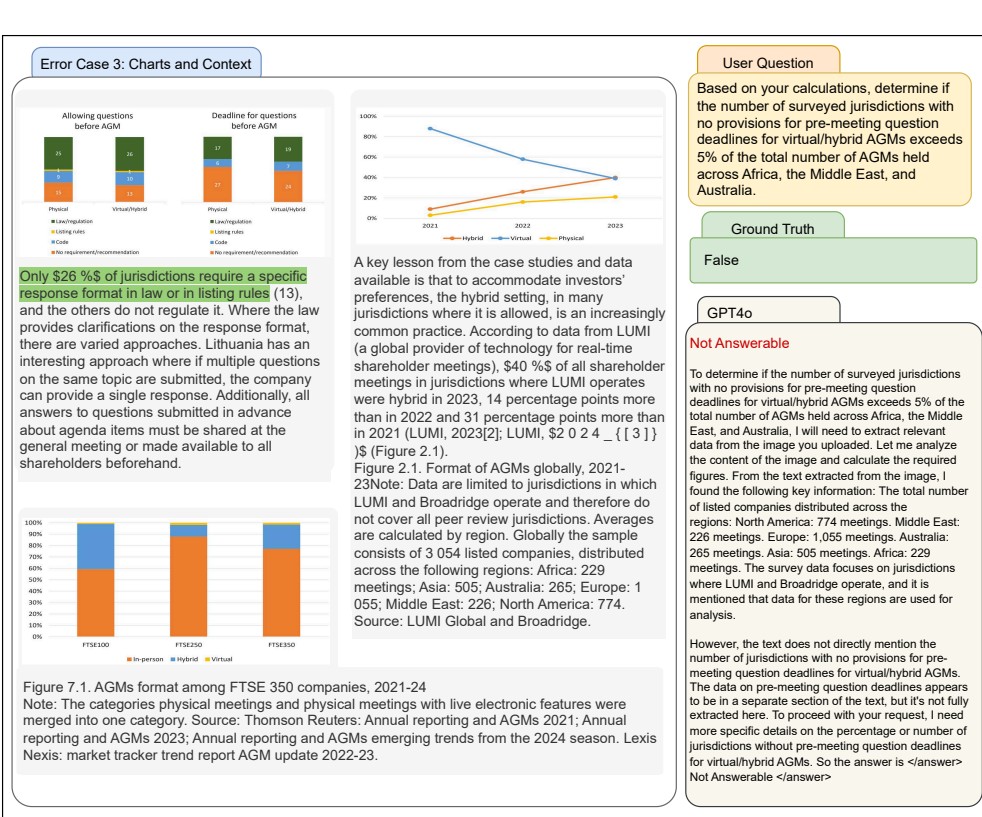

Figure 35: Error case 3. In this case, GPT-4o focused only on the chart and overlooked the accompanying context. The highlighted section in the context implicitly states the total value, and the model could only answer correctly by integrating information from both the document context and the chart itself. This highlights how our benchmark better reflects real-world document understanding scenarios.

## G   Additional Experimental Analysis

To provide a comprehensive diagnosis of model capabilities and validate the unique challenges posed by ChartNexus, we conducted a series of fine-grained ablation studies and stratified analyses.

### G.1   Performance by Chart Type

To address concerns regarding the distribution of chart types and to identify specific model " failure modes," we performed a category-wise performance breakdown across all chart types present in the benchmark. As shown in Table 8, while models perform relatively well on common 2D charts (e.g., Bar, Line, Pie), we observe a significant performance degradation on complex spatial visualizations.

Notably, all evaluated models exhibit a "performance cliff" on **3D charts**, with the best-performing model (Qwen2.5-VL-32B) achieving only 36.1% accuracy, compared to 65.4% on Bar charts. This universal deficit highlights a critical limitation in current Vision-Language Pretraining regarding spatial projection and depth perception.

Table 8: Fine-grained performance breakdown by chart type. Note the significant performance drop on **3D charts** across all models compared to standard types, revealing a boundary in current spatial reasoning capabilities.

| Model | Bar | Line | Pie | Table | Scatter | Tree | Radar | Area | Sunburst | Graph | Boxplot | Sankey | Heatmap | 3D | Candle | Funnel | Other |
|---|---|---|---|---|---|---|---|---|---|---|---|---|---|---|---|---|---|
| GPT-4o | 63.7 | 62.5 | 74.8 | 70.2 | 63.9 | 69.9 | 60.0 | 56.6 | 70.5 | 66.7 | 62.0 | 100.0 | 64.4 | **33.3** | 63.6 | 76.9 | 64.9 |
| Qwen2.5-VL-32B | 65.4 | 62.9 | 80.2 | 73.2 | 56.4 | 69.1 | 50.0 | 64.2 | 64.7 | 66.7 | 62.5 | 73.6 | 65.6 | **36.1** | 54.5 | 69.2 | 64.7 |
| InternVL3-38B | 63.1 | 63.8 | 65.2 | 77.2 | 81.6 | 66.6 | 65.0 | 66.6 | 64.7 | 46.7 | 68.6 | 100.0 | 85.0 | **33.3** | 63.6 | 84.6 | 75.0 |
| GLM-4.1V-9B | 66.7 | 65.7 | 77.3 | 79.0 | 66.5 | 70.2 | 50.0 | 59.5 | 58.8 | 66.7 | 37.5 | 84.2 | 55.4 | **17.2** | 63.6 | 61.5 | 64.6 |
| InternVL3-14B | 58.3 | 58.7 | 79.1 | 80.1 | 36.8 | 50.6 | 55.0 | 63.6 | 66.7 | 53.3 | 23.5 | 89.4 | 20.0 | **20.0** | 45.4 | 61.5 | 40.2 |
| Qwen2.5-VL-7B | 46.8 | 44.5 | 48.6 | 60.7 | 38.4 | 51.4 | 45.0 | 45.2 | 58.8 | 33.3 | 25.0 | 89.4 | 44.3 | **16.8** | 27.2 | 53.8 | 48.9 |

### G.2   Impact of Textual Context

To quantify the role of document context in multi-chart reasoning, we conducted an ablation study where all textual context (captions and related paragraphs) was removed, leaving only the chart images.

As shown in Table 9, performance drops significantly in the "No-Context" setting compared to the full benchmark (e.g., Qwen2.5-VL-7B drops from 46.67% to 15.1% in Open-Ended Vocabulary tasks). This quantitative "Context Gap" confirms that visual perception alone is insufficient for ChartNexus tasks, which require the model to use text as a semantic bridge to interpret and connect visual data.

Table 9: Model performance in the **No-Context** ablation setting. Comparing these results with the main table reveals the critical dependency on cross-modal grounding.

| Models | Bool | Approx. Value | Multi-Choice | Open-Ended (Vocab) | Open-Ended (Sent.) |
|---|---|---|---|---|---|
| Qwen2.5-VL-32B | 52.3 | 42.7 | 60.1 | 30.5 | 68.1 |
| InternVL3-38B | 53.1 | 54.3 | 51.5 | 21.2 | 57.7 |
| InternVL3-14B | 43.0 | 37.8 | 53.4 | 25.2 | 56.4 |
| GLM-4.1V-9B | 41.7 | 40.6 | 54.6 | 27.8 | 55.5 |
| Qwen2.5-VL-7B | 31.7 | 34.5 | 44.6 | 15.1 | 43.2 |

### G.3   Composite Subplots vs. Discrete Images

We further analyzed whether the difficulty in multi-chart reasoning stems from logical complexity (reasoning across files) or visual density (parsing subplots). We categorized samples into **Single image containing multiple subplots** and **Multiple discrete image files**.

Table 10 shows that models like InternVL3-14B suffer a massive drop on subplot samples (37.7%) compared to discrete images (61.2%), indicating a bottleneck in **visual resolution** or encoder capacity when processing dense composite figures. Larger models (e.g., InternVL3-38B) show robustness across both settings.

Table 10: Performance comparison between composite images (Subplots) and discrete images. The discrepancy in smaller models highlights visual resolution bottlenecks.

| Models | Multi-Charts with Subplots | Multi-Charts w/o Subplots |
|---|---|---|
| InternVL3-38B | 64.2 | 64.1 |
| Qwen2.5-VL-32B | 60.3 | 65.7 |
| GLM-4.1V-9B | 61.5 | 68.2 |
| InternVL3-14B | 37.7 | 61.2 |
| Qwen2.5-VL-7B | 46.5 | 46.3 |

### G.4 HALLUCINATION ANALYSIS: BOOLEAN VS. MULTI-CHOICE

To justify the inclusion of a distinct "Judgment" (Boolean) category, we analyzed the models' refusal capabilities. We define a *Hallucination* as the model providing a specific answer when the ground truth is "Unanswerable" (False Negative). Conversely, a True Positive (TP) occurs when the model correctly identifies the question as unanswerable.

Table 11 illustrates a strong "Selection Bias" inherent in the Multiple-Choice format. Models like GLM-4.1V-9B, InternVL3-14B, and Qwen2.5-VL-32B achieved 0 successful refusals ($TP = 0$) in the Multi-Choice setting, effectively hallucinating an answer in 100% of unanswerable cases. Even GPT-4o showed a significantly higher hallucination rate in Multiple-Choice compared to Judgment tasks.

This dissociation demonstrates that the "Judgment" format effectively exposes a model's latent fact-checking capabilities, which are often overridden by the structural bias of multiple-choice prompts. Therefore, the Boolean category serves as an indispensable diagnostic tool for evaluating faithfulness and refusal capability.

Table 11: Comparison of Hallucination Rates on Unanswerable Questions. **TP** (True Positive) indicates a correct refusal (predicting "Unanswerable"); **FN** (False Negative) indicates a hallucination (predicting an option/value). The Judgment format significantly outperforms Multi-Choice in eliciting correct refusals.

| Model | Judgment (Boolean) | | | Multiple-Choice | | |
|---|---|---|---|---|---|---|
| | TP | FN (Hallucination) | F1 Score | TP | FN (Hallucination) | F1 Score |
| GPT-4o | 7 | 5 | 0.333 | 3 | 13 | 0.124 |
| Qwen2.5-VL-7B | 6 | 50 | 0.176 | 1 | 15 | 0.047 |
| GLM-4.1V-9B | 3 | 9 | 0.188 | 0 | 16 | 0.000 |
| InternVL3-14B | 2 | 10 | 0.118 | 0 | 16 | 0.000 |
| Qwen2.5-VL-32B | 2 | 10 | 0.182 | 0 | 16 | 0.000 |
| InternVL3-38B | 10 | 2 | 0.589 | 2 | 14 | 0.181 |

## H  Reliability and Validity Checks

### H.1  Human Performance Baseline

To quantify the gap between current MLLMs and human capabilities, we established an explicit human performance baseline. Two expert annotators (graduate students) were recruited to evaluate a stratified sample of 30 instances per question type.

As shown in Table 12, the average human accuracy is approximately **89.1%**, which significantly outperforms current SOTA models. This confirms that while the benchmark tasks are solvable, they remain challenging even for humans due to the complexity of real-world data.

Table 12: Human performance baseline across different task categories.

| | Bool | Approx. Value | Multi-Choice | Open-Ended (Voc) | Open-Ended (Sent) |
|---|---|---|---|---|---|
| **Human** | 93.3 | 90.0 | 83.3 | 93.3 | 85.7 |

### H.2  Sensitivity to Evaluation Prompt Language

Given the multilingual nature of the community, we assessed whether the language of the evaluation prompt (Chinese vs. English) affects the scoring of the SEAT metric. We compared the original Chinese prompts with professionally translated English prompts.

Table 13 shows a minor absolute performance shift (approx. 1.5% - 3.5%) but, crucially, the **relative ranking of models remains identical**. This confirms the robustness of our benchmark's conclusions regardless of the evaluator's prompt language.

Table 13: Ablation study on SEAT evaluation prompt language.

| Models | Chinese Prompt | English Prompt |
|---|---|---|
| Qwen2.5-VL-32B | 72.67 | 70.28 |
| GLM-4.1V-9B | 68.77 | 65.29 |
| InternVL3-38B | 58.96 | 56.39 |
| InternVL3-14B | 56.25 | 52.98 |
| Qwen2.5-VL-7B | 49.49 | 48.01 |

### H.3  Reliability of Automated Evaluation

**Human-Model Alignment Study.** To address concerns regarding the reliability of Qwen3-32B as an automated judge, we conducted a human verification study on a stratified sample of 250 instances (50 per question type).

As shown in Table 14, the judge achieves near-perfect alignment (98%–100%) for objective tasks (Boolean, Multi-Choice, Approximate Value). The 6.5% overall misalignment is concentrated in Open-Ended tasks, primarily due to linguistic ambiguity in SEAT decomposition rather than systematic bias. This confirms Qwen3-32B is a reliable proxy for human evaluation.

Table 14: Human-Model Agreement Rates by Question Type.

| Type | Bool | Approx. Value | Multi-Choice | Open (Vocab) | Open (Sent) |
|---|---|---|---|---|---|
| Agreement (%) | 100 | 98 | 100 | 92 | 86 |

**Robustness of Automated Evaluation Across Judges**

To ensure our rankings are not artifacts of a specific judge model's bias, we conducted an extensive comparative study on the "Open-Ended (sentence)" category using the SEAT method. We employed `Deepseek-chat` and `GPT-4o` as independent judges, comparing their scoring distributions and resulting rankings against our original judge (`Qwen3-32B`) and human verification.

For human verification, two experts evaluated a stratified sample of 30 responses per model, whereas the automated judges evaluated the full benchmark. As shown in Table 15, the results reveal a high degree of consistency in model rankings across diverse judges. While absolute scores vary—for instance, `GPT-4o` tends to be stricter, assigning lower scores across the board—the relative ordering of the evaluated models remains stable.

Table 15: Comparison of Model Performance Scores in the "Open-Ended (sentence)" Category under Different Evaluators. Despite variations in absolute scores, the relative ranking of models remains consistent.

| Judge Model | Qwen2.5-VL-7B | GLM-4.1V-9B | InternVL3-14B | Qwen2.5-VL-32B | InternVL3-38B |
|---|---|---|---|---|---|
| Human (Sampled) | 47.2 | 52.5 | 50.4 | 67.2 | 57.3 |
| Qwen3-32B (Ours) | 49.5 | 68.8 | 56.3 | 72.7 | 60.0 |
| Deepseek-chat | 44.8 | 64.7 | 59.1 | 70.9 | 61.0 |
| GPT-4o | 39.3 | 60.1 | 56.1 | 61.9 | 57.8 |

