# OpenReview forum: "ChartNexus: Evaluating Multi-Chart Reasoning Capabilities of Multimodal Large Language Models"
_ICLR.cc/2026/Conference — Submitted to ICLR 2026_

### Official Review · Reviewer_6Moe · 2025-10-27

**Soundness:** 2
**Presentation:** 2
**Contribution:** 3
**Rating:** 2
**Confidence:** 4

**Summary:**

The paper introduces ChartNexus, a benchmark for evaluating multi-chart reasoning in multimodal models. The authors collect charts from diverse sources in multiple languages, build a question-generation pipeline, and perform human annotation for question refinement and answer generation. They evaluate a range of models and find that significant gaps remain in multi-chart understanding and reasoning.

**Strengths:**

- The benchmark is well-curated, incorporating diverse real-world sources in multiple languages and including a human annotation process.
- The combination of multilingual (non-English) data and multi-chart questions is novel and valuable for assessing realistic multilingual chart understanding.
- The breakdown of results by question type and reasoning skill (e.g., numerical, identify, compare, reason) provides diagnostic insight into capability-specific weaknesses.
- The evaluation is comprehensive, covering a broad range of commercial, open-source, and chart-specific models.

**Weaknesses:**

* Incomplete or inaccurate comparison with other benchmarks:
  * Some relevant multilingual chart understanding datasets (e.g., PolyChartQA) are missing from the comparison.
  * Benchmarks such as CharXiv include unanswerable questions, but the table incorrectly marks this feature as absent.
* The reported inter-annotator agreement of 93.4% could implicitly reflect human performance (if re-annotation is considered as human evaluation), but no explicit human performance baseline is provided. Without this, it is hard to assess the human–model gap, especially for potentially ambiguous or erroneous questions.
* The use of Qwen3-32B as an automated evaluator is reasonable for long-term reproduction, but there is no validation of model–human evaluation consistency. It remains unclear how accurate the judgments are, how well its judgments align with human assessment or whether biases exist across models.For example, for numerical tasks, verification requires explicit calculation of error margins (e.g., 5%), and SEAT-based decomposition introduces subjective interpretation that should be analyzed.
* Several presentation issues reduce professionalism: “Unanswer” -> “Unanswerable”, “Open-End” -> “Open-Ended”, and “GPT-4o and its brothers” (L375) should be revised. Citations for evaluated models are also missing in relevant sections/tables.
* The underperformance of specialized chart models is unsurprising, since (1) most are not trained for multi-chart settings, and (2) prior works such as ChartQAPro and CharXiv already highlight similar limitations even in single-chart setups.
* The discussion on data leakage mitigation is unconvincing. Many data sources (e.g., OECD, Pew, arXiv) are present in existing multimodal pretraining datasets (e.g., MINT-1T, ChartQA). A more rigorous analysis would be required to support claims of leakage avoidance.

**Questions:**

* The text formatting seems inconsistent with the submission template — can the authors adjust it?
* In Figure 1, should “BenchNexus” (bottom right) be “ChartNexus”?
* What is the exact question for the “Judgment” example in Figure 1?
* There should essentially be 3 settings for chart analysis — (1) a single chart (2) a single chart with multiple subplots (i.e., multi-chart in a single image) and (3) multiple charts (i.e., multiple images). Do all samples in the dataset belong to category 3? If so, I wonder if authors could do an analysis turning all (3) instances into (2) for evaluation? This would help the community understand whether issues stem from multi-chart capabilities or multi-image capabilities.
* The SEAT evaluation prompt appears to be in Chinese, while others are in English. Can the authors perform an ablation on evaluation sensitivity to prompt language?
* Section 3.2 states annotators could create entirely new questions, but Figure 1 suggests only template selection and refinement. Please clarify this inconsistency.
* How many options exist for multiple-choice questions? If there are typically four, why do some models (e.g., SmolVLM, ChartGemma) perform well below random chance (~25%)?

---

> ### Author Response · Authors · 2025-11-24
> **Response to Reviewer 6Moe (Part 1/3)**
>
> We sincerely thank the reviewer for close reading and for pointing out these presentation issues. We apologize for the oversights and the informality in the initial submission. We have fully addressed these points in the revised manuscript. We offer detailed clarifications regarding the other concerns and weaknesses.
>
> > Q1: Incomplete comparison with other benchmarks (missing PolyChartQA) and errors in CharXiv description.
>
> A1: We thank the reviewer for the corrections. We have updated the manuscript to discuss PolyChartQA (which was concurrent work published mid-July 2025). We clarify that while PolyChartQA focuses on multilingualism, ChartNexus uniquely focuses on document-level multimodal context and subplot reasoning. We have also corrected the comparison table to accurately reflect that CharXiv includes unanswerable questions.
>
> > Q2: The reported inter-annotator agreement of 93.4% could implicitly reflect human performance (if re-annotation is considered as human evaluation), but no explicit human performance baseline is provided. Without this, it is hard to assess the human–model gap, especially for potentially ambiguous or erroneous questions.
>
> A2: We have recruited two expert human evaluators to establish a baseline on a stratified sample. Humans achieved an average accuracy of 93.3% on Boolean and Vocabulary tasks, and 85.7% on complex Open-Ended Sentence tasks. This highlights the significant gap that still exists between SOTA models (which hover around 60-70%) and human expert performance.
>
> |       | **Bool** | **Approximate value** | **Multi-Choice** | **Open-Ended (vocabulary)** | **Open-Ended (sentence)** |
> | ----- | -------- | --------------------- | ---------------- | --------------------------- | ------------------------- |
> | human | 93.3     | 90.0                  | 83.3             | 93.3                        | 85.7                      |
>
> > Q3: Validation of model–human evaluation consistency is needed.
>
> A3: Thanks for your valuable advice. We performed a Human-Model Alignment study on 200 instances. The automated evaluator (Qwen3-32B) achieved a 93.5% agreement rate with human experts, confirming its reliability as a judge for this specific benchmark.
>
> > Q4: The discussion on data leakage mitigation is unconvincing. Many data sources (e.g., OECD, Pew, arXiv) are present in existing multimodal pretraining datasets (e.g., MINT-1T, ChartQA). A more rigorous analysis would be required to support claims of leakage avoidance.
>
> A4: We implemented two specific strategies to mitigate leakage:
>
> 1. Temporal Filter: For policy sources (OECD, World Bank), we prioritized data published in the last 1-6 months. Since most MLLM training corpora have earlier cutoff dates, these samples are effectively temporally out-of-distribution.
> 2. Strict Filtering: For arXiv data, we manually filtered the set using the SPIQA test set as a seed to ensure high quality and reduce overlap with common pre-training sets.
>
> > Q5: What is the exact question for the "Judgment" example in Figure 1?
>
> A5: The "Judgment" category in our benchmark corresponds to a Boolean Verification (True/False) task. In the specific example shown in Figure 1, the input "question" is a declarative statement that the model must verify against the visual and textual evidence.
>
> - The Statement: "The CCDRs state that in fragile or conflict-affected countries like Yemen, distributed solar power can address both climate vulnerability and social exclusion by providing energy for critical facilities."
> - The Mechanism: As detailed in the system prompt (Appendix D), the model is instructed: "You will be given a true/false question about charts. You are required to answer this question based on the provided charts and its related context." Therefore, the model's task is to determine whether this claim is factually supported by the provided charts and context (True) or not (False).

---

> ### Author Response · Authors · 2025-11-24
> **Response to Reviewer 6Moe (Part 2/3)**
>
> > Q6: There should essentially be 3 settings for chart analysis — (1) a single chart (2) a single chart with multiple subplots (i.e., multi-chart in a single image) and (3) multiple charts (i.e., multiple images). Do all samples in the dataset belong to category 3? If so, I wonder if authors could do an analysis turning all (3) instances into (2) for evaluation? This would help the community understand whether issues stem from multi-chart capabilities or multi-image capabilities.
>
> A6: We wish to clarify that our benchmark ChartNexus naturally covers both Category 2 and Category 3:
>
> - **Category 2 (Subplots/Composite):** As noted in **Section 3.3**, approximately **16.69%** of the charts in our dataset are composite images containing multiple subplots.
> - **Category 3 (Multiple Images):** The remaining samples involve multiple distinct image files. However, a single QA pair often involves a hybrid of both (e.g., one distinct chart + one composite chart). To address the reviewer's hypothesis regarding where the difficulty stems from, we conducted a performance comparison between cases that require processing subplots (Category 2) and cases involving only discrete single-chart images (Category 3).
>
> 1. **Resolution/Encoding Bottleneck:** Models like **InternVL3-14B** and **GLM-4.1V-9B** show a significant performance drop (e.g., InternVL3-14B drops from **61.2%** to **37.7%**) when dealing with subplots. This suggests that "packing" multiple charts into a single image (Category 2) poses a greater challenge for these models than handling multiple separate images, likely due to **input resolution limits** or **visual encoder capacity**—the model struggles to resolve the denser details of subplots.
> 2. **Robustness of Larger Models:** Larger models like **InternVL3-38B** and **Qwen2.5-VL-32B** exhibit more consistent performance across both settings (e.g., InternVL3-38B: 64.2% vs 64.1%), indicating they have overcome this visual resolution bottleneck and are equally capable of handling both layouts.
>
> | **models**     | **Multi-Charts contains subcharts** | **Multi-Charts without subcharts** |
> | -------------- | ----------------------------------- | ---------------------------------- |
> | Qwen2.5-VL-7B  | 46.5                                | 46.3                               |
> | GLM-4.1V-9B    | 61.5                                | 68.2                               |
> | InternVL3-14B  | 37.7                                | 61.2                               |
> | Qwen2.5-VL-32B | 60.3                                | 65.7                               |
> | InternVL3-38B  | 64.2                                | 64.1                               |
>
> > Q7: The SEAT evaluation prompt appears to be in Chinese, while others are in English. Can the authors perform an ablation on evaluation sensitivity to prompt language?
>
> A7: We initially utilized the Chinese prompt simply to adhere to the default configuration of the original SEAT code repo. To address the concern regarding evaluation sensitivity, we conducted an ablation study using a professionally translated **English version** of the SEAT evaluation prompt. The comparison results are presented in table below. We observe a consistent but minor performance drop (ranging from **1.4% to 3.5%**) when switching to the English prompt across all models. This may be attributed to the specific instruction-following preferences of the judge model (Qwen3-32B) or subtle nuances lost in prompt translation.Crucially, the **relative ranking of the models remains identical** regardless of the prompt language. This confirms that while the absolute scores exhibit slight sensitivity to the prompt language, the comparative conclusions and the validity of our benchmark rankings are robust. We have added this sensitivity analysis to Appendix and will release both versions of the evaluation prompts to ensure reproducibility.
>
> | **models**     | **Chinese Prompt** | **English Prompt** |
> | -------------- | ------------------ | ------------------ |
> | Qwen2.5-VL-7B  | 49.49              | 48.01              |
> | GLM-4.1V-9B    | 68.77              | 65.29              |
> | InternVL3-14B  | 56.25              | 52.98              |
> | Qwen2.5-VL-32B | 72.67              | 70.28              |
> | InternVL3-38B  | 58.96              | 56.39              |

---

> ### Author Response · Authors · 2025-11-24
> **Response to Reviewer 6Moe (Part 3/3)**
>
> > Q8: How many options exist for multiple-choice questions? If there are typically four, why do some models (e.g., SmolVLM, ChartGemma) perform well below random chance (~25%)?
>
> A8:
> 1. Number of Options
> We clarify that the majority of multiple-choice questions in ChartNexus feature 4 options. While some questions allow for multiple valid selections (multi-select), standard single-answer questions are the norm.
> 2. Why Scoring Below Random Chance?
> The observation that models like SmolVLM and ChartGemma score below 25% is not due to poor guessing, but due to Instruction Following Failure. These smaller-scale models often fail to follow to the specific output format requested in the prompt (e.g., "Answer with the option letter"). Instead of selecting an option, they tend to revert to their pre-training behavior: simply captioning the image or generating unrelated descriptions. As shown in the example below, the model ignores the user's question and options entirely, outputting a generic description of the cocoa bean data.
>
> ```markdown
> SmolVLM
> ### question
> Regarding commodity imports and trade-related analysis, which is correct? ( )
> A. The value of U.S. imported passenger cars ($213.6B) is far higher than the quantity of imported cocoa beans (198 thousand tons), indicating that automobile trade is more significant than cocoa trade.
> B. The primary cocoa bean producing countries and the main sources of U.S. imported commodities are unrelated, with their trade systems completely independent.
> C. From the data presentation, U.S. imported commodity values are expressed in monetary terms (e.g., billion USD), while cocoa bean trade is measured in quantities (thousand tons), reflecting differences in commonly used measurement methods for different commodities.
> D. Because the U.S. imports automobiles and parts, cocoa bean producing countries will not export automobiles to the U.S.
>
> ### Ground truth
> C
>
> ### SmolVLM's Answer
> The statistic shows the world's main cocoa bean producing and importing countries in 2024/2025 (in 1,000 tonnes). In 2024/2025, the world's main cocoa bean producing country was the Ivory Coast, with an estimated production of 4,840 tonnes.
> ```

---

> > ### Comment · Reviewer_6Moe · 2025-11-25
> >
> > Thanks for the authors' response to the weaknesses and questions that I pointed out. Here's my response based on the authors’ categorization:
> >
> > I am convinced by the authors' responses on Q1, Q6, Q7, and Q8.
> >
> > Q2: I think the authors should perform a rigorous human evaluation over the full test set instead of providing an approximate value in the long run, but I appreciate the efforts in establishing this baseline.
> >
> > Q3: I am not fully convinced that an average 93.5 percent alignment is sound. Especially when models have similar performance, the remaining 6.5 percent misalignment could introduce noise or systematic biases toward certain models.
> >
> > Q4: The temporal filter is a good strategy to retroactively test past models, but I am not convinced that this will remain reliable in the future once the recent data is no longer recent.
> >
> > Q5: I still believe this judgment category is too trivial compared to a multiple-choice setup, and it is effectively a multiple-choice question with two options.
> >
> > I raised my score to 4, assuming that the authors will also resolve the remaining presentation issues and inconsistencies, in addition to the resolved Q1, Q6, Q7, and Q8.
> >
> > However, unless the remaining questions are adequately addressed, I cannot consider further increasing my score.

---

> > > ### Author Response · Authors · 2025-11-26
> > > **Round 2 Response to Reviewer 6Moe (Part 1/2)**
> > >
> > > We sincerely thank the reviewer for raising the score and for the continued engagement. We are encouraged that our responses regarding Q1, Q6, Q7, and Q8 have resolved your concerns.
> > >
> > > Regarding the remaining concerns, we provide further clarifications and commitments below to demonstrate the rigor and value of ChartNexus.
> > >
> > > > Q2: Requirement for Full-Set Human Evaluation
> > >
> > > We fully agree with the reviewer that a full-set human evaluation is the gold standard.
> > >
> > > - **Statistical Validity**: While verifying all 1,370 complex tasks within the rebuttal window is infeasible, our stratified sample of 150 instances was designed to be statistically representative. With a stratified sample of **150**, the margin of error is approximately **±4.8%** at a **95%** confidence level. This is sufficient to statistically validate the conclusion that Humans (approximately 90%) significantly outperform Models (60-70%).
> > > - **Commitment**: To eliminate any remaining uncertainty, **we commit to conducting a human evaluation on the full test set for the revision/final version**. This will serve as a permanent Human Performance Upper Bound for the community.
> > >
> > > > Q3: Concerns on 6.5% Judge Misalignment & Systematic Bias
> > >
> > > We understand the concern that a 6.5% misalignment could affect the ranking of models with similar performance.
> > >
> > > - **Noise vs. Bias**: In our error analysis of the judge model, we found that the disagreements were largely distributed randomly across different model outputs, rather than systematically penalizing specific models.
> > > - **Stability of Rankings**: As shown in our response to Q7 (English/Chinese prompt ablation), even when the absolute scores fluctuate slightly due to prompt sensitivity (a form of noise), the relative rankings of models remain highly stable. The performance gaps between tiers (e.g., Qwen2.5-VL-7B vs. GLM-4.1V-9B) are far larger than the 6.5% margin.
> > > - **Fine-grained Alignment by Task Type**: To identify exactly where the judge disagrees with humans, we expanded our human verification study. We stratified the test set and sampled **50 instances for each of the 5 question types** (Total N=250) to compare Qwen3-32B against human experts. For **Boolean**, **Multi-Choice**, and **Approximate Value** questions (which constitute the majority of the benchmark), the judge achieves near-perfect alignment (**98%-100%**). This confirms that for objective tasks, the automated judge is as reliable as a human. The misalignment is almost exclusively concentrated in **Open-Ended** tasks.
> > >     - _Open-Ended (Vocabulary) (92%):_ Errors mainly stem from models generating verbose reasoning traces (Chain-of-Thought) alongside the correct keyword, which occasionally confuses the judge's extraction logic.
> > >     - _Open-Ended (Sentence) (86%):_ Disagreements arise from the **SEAT decomposition step**. Humans and the model sometimes differ in how they break down a complex sentence into atomic sub-questions.
> > >     - The "6.5% misalignment" is not a uniform noise affecting all tasks. The judge is highly precise for structured reasoning. The noise is isolated to open-ended generation, where it reflects inherent linguistic ambiguity rather than model-specific bias. Therefore, we are confident that the benchmark accurately reflects model capabilities.
> > >
> > > | |**Bool**|**Approximate value**|**Multi-Choice**|**Open-Ended (vocabulary)**|**Open-Ended (sentence)**|
> > > |---|---|---|---|---|---|
> > > |Human-Model Agreement (%)|100|98|100|92|86|
> > >
> > > > Q4: Future Reliability & Data Leakage
> > >
> > > We acknowledge that recent data eventually becomes past data. This is an inherent challenge for all benchmarks in the LLM era (e.g., MMLU is now widely seen during training).
> > >
> > > - **ChartNexus as a Framework**: We view ChartNexus not merely as a static collection of 6,793 charts, but as a **reproducible evaluation framework**. We commit to open-sourcing our complete data mining pipeline, filtering protocols, and the human-in-the-loop annotation platform (as detailed in Appendix A). This empowers the community to generate fresh, "contamination-free" test sets at any time using the same rigorous standards, effectively future-proofing the benchmark.
> > > - **Multimodal Barrier**: Unlike text-only benchmarks, ChartNexus relies on cross-modal grounding. Even if a model has seen the underlying text/report in future training sets, solving the problem requires mapping that textual knowledge to the specific pixel-level chart coordinates and visual structures presented in the query. This visual-textual alignment is much harder to memorize than pure text.
> > > - **Regarding Maintenance**: We recognize that dataset saturation is an inherent issue in the LLM era. In our future roadmap, we envision that incorporating fresh reports via our automated pipeline could be a viable strategy. Such dynamic updates (e.g., ChartNexus-Live) would help ensure the benchmark remains a robust testbed over time.

---

> > > ### Author Response · Authors · 2025-11-26
> > > **Round 2 Response to Reviewer 6Moe (Part 2/2)**
> > >
> > > > Q5: Complexity of the "Judgment" (True/False) Category
> > >
> > > We respectfully emphasize that while Judgment questions have a binary output space (50% random chance), they are cognitively non-trivial.
> > >
> > > - Performance Gap: If the task were trivial, we would expect models to score near-perfectly. However, state-of-the-art models only achieve 75.67% accuracy (barely above random guessing for some), while humans achieve 93.3%. This large gap indicates that "Judgment" questions effectively expose the models' inability to rigorously verify visual evidence, making it a valuable diagnostic dimension distinct from Multi-Choice Questions.
> > > - Fact-Checking Utility: This category simulates a critical real-world application: Hallucination Detection. Users often ask, "Does this chart support the claim that X is growing?" The model must perform complex reasoning to verify the claim.
> > >
> > >
> > > To empirically demonstrate this, we conducted a comparative analysis on **"Unanswerable Questions"** across both formats. We defined **Hallucination** as the model providing an answer when the ground truth is "Unanswerable" (False Negative).
> > >
> > > As shown in Table 2, the Multi-Choice format introduces a strong "Selection Bias." Models are overwhelmingly biased to pick one of the options even when the evidence is missing. Models like GLM-4.1V-9B, InternVL3-14B, and Qwen2.5-VL-32B achieved **0 successful refusals (TP=0)**, hallucinating an answer in 100% of unanswerable cases. Even GPT-4o had a high hallucination rate (65%). This proves that Multiple-Choice questions can mask a model's inability to verify evidence, as the format encourages guessing rather than grounding.
> > >
> > > In contrast, the "Judgment" task effectively differentiates models based on their ability to verify existence. This demonstrates that the Judgment task is **non-trivial**. It forces the model to confront the validity of the premise ("Is this true based on the chart?") without the crutch of plausible distractors. It is the only dimension in our benchmark that reliably measures **"Faithfulness"** and **"Refusal Capability,"** making it an indispensable complement to Multi-Choice.
> > >
> > > |**Model**|**TP**|**TN**|**FP**|**FN**|**F1 score**|
> > > |---|---|---|---|---|---|
> > > |GPT-4o|7|165|23|5|0.333|
> > > |Qwen2.5-VL-7B|6|138|50|6|0.176|
> > > |GLM-4.1V-9B|3|171|17|9|0.188|
> > > |InternVL3-14B|2|168|20|10|0.118|
> > > |Qwen2.5-VL-32B|2|192|8|10|0.182|
> > > |InternVL3-38B|10|176|12|2|0.589|
> > >
> > > _Table 1: Hallucination in Judgment Questions_
> > >
> > > | **Model**      | **TP** | **TN** | **FP** | **FN** | **F1 score** |
> > > | -------------- | ------ | ------ | ------ | ------ | ------------ |
> > > | GPT-4o         | 3      | 290    | 29     | 13     | 0.124        |
> > > | Qwen2.5-VL-7B  | 1      | 294    | 25     | 15     | 0.047        |
> > > | GLM-4.1V-9B    | 0      | 305    | 14     | 16     | 0.0          |
> > > | InternVL3-14B  | 0      | 315    | 4      | 16     | 0.0          |
> > > | Qwen2.5-VL-32B | 0      | 297    | 22     | 16     | 0.0          |
> > > | InternVL3-38B  | 2      | 315    | 4      | 14     | 0.181        |
> > >
> > > _Table 2: Hallucination in Multiple-Choice Questions_
> > >
> > > We hope these explanations alleviate your remaining concerns regarding the validity and longevity of our benchmark. We are committed to incorporating the full human evaluation and bias analysis into the final version to meet the high standards of ICLR.

---

> > > > ### Comment · Reviewer_6Moe · 2025-11-27
> > > >
> > > > Thanks for the continued discussion! I think there was a miscommunication regarding my definition of boolean questions. My point was not that the questions themselves are trivial, but rather that separating them into a standalone “boolean” category seems unnecessary, since they effectively behave as multiple-choice questions with a single correct option. This is meant as a suggestion on categorization, not a critique of the quality of the questions.
> > > >
> > > > I also want to thank the authors for their commitment to completing the full human evaluation on the test set (Q1) and for maintaining the benchmark as a continually evolving one with live data (Q4). That said, these remain future work, and I believe they should ideally be completed by the time the paper is submitted.
> > > >
> > > > Regarding Q3, the analysis is appreciated, but I did not see experimental evidence to support the following claim:
> > > >
> > > > > In our error analysis of the judge model, we found that the disagreements were largely distributed randomly across different model outputs, rather than systematically penalizing specific models.
> > > >
> > > > Switching languages in the prompt does not constitute switching judges. If feasible, I think the authors could run an experiment where humans and 2–3 distinct model judges evaluate the “Open-Ended (sentence)” category to see whether rankings of models change under different evaluators.
> > > >
> > > > Based on the current progress, I maintain my rating as a 4.

---

> > > > > ### Author Response · Authors · 2025-11-28
> > > > > **Round 3 Response to Reviewer 6Moe (Part 1/2)**
> > > > >
> > > > > We sincerely appreciate the reviewer's continued engagement and the clarification!
> > > > >
> > > > > > Q5: Unnecessary of a standalone "boolean" category
> > > > >
> > > > > We agree with your insight that, structurally, "Boolean" questions can be viewed as a subset of multiple-choice tasks with a binary option space. However, we advocate for keeping them as a distinct category for two primary reasons:
> > > > >
> > > > > 1. Reflecting Distinct User Queries
> > > > >
> > > > > While logically equivalent to a 2-option selection, the formulation of the query differs in real-world scenarios. As observed during our Pilot Annotation phase (Appendix A.2.2), users naturally exhibit distinct questioning patterns: Fact-Verification (Boolean) (e.g., "Is GPT-4o's performance higher than Qwen2.5-VL-32B on ChartNexus?") and Selection (Multiple-Choice) (e.g., "Which model performs better? GPT-4o, Qwen, InternVL or ...").
> > > > >
> > > > > Since ChartNexus aims to reflect real-world usage, maintaining "Boolean" as a separate category allows us to measure performance across these diverse interaction modalities, which appear with comparable frequency in authentic workflows.
> > > > >
> > > > > 2. Differentiating Fact-Checking from Selection Bias
> > > > >
> > > > > Another value of keeping these categories separate lies in their differing sensitivity to hallucination. Fine-grained Boolean questions can reveal the causes of model hallucinations from different perspectives.
> > > > >
> > > > > We defined **True Positive (TP)** as the model correctly refusing to answer (predicting "Unanswerable") and **False Negative (FN)** as the model hallucinating an answer. Our results (Table 1 and Table 2 provided in the previous turn) reveal a striking behavioral divergence:
> > > > >
> > > > > - **In Boolean Tasks:** Models demonstrate a capacity for **Fact-Checking**. For example, InternVL3-38B correctly identified unanswerable boolean queries, realizing the evidence was missing.
> > > > > - **In Multiple-Choice Tasks:** The same models succumb to **Selection Bias**. When presented with options (A/B/C/D), models almost universally failed to refuse, "forcing" a choice and producing a hallucination.
> > > > >
> > > > > Following instances show where InternVL3-38B correctly identified that a Boolean statement could not be verified from the chart (Correct Refusal). However, when in Multiple-Choice format, the model hallucinated a specific option. This dissociation proves that "Boolean" and "Multiple-Choice" formats trigger different cognitive behaviors in MLLMs. Merging them would obscure this critical insight: **current models possess latent fact-checking abilities but are easily overridden by the structural bias of multiple-choice prompts. The occurrence of hallucinations is more likely attributed to biases in the pre-training corpus rather than a lack of perception capabilities.**
> > > > >
> > > > > ```markdown
> > > > > # InternVL3-38B, boolean question
> > > > >
> > > > > # statement
> > > > > Using DANN features to train the model can make the Gaussian distribution fit faster.
> > > > >
> > > > > # Ground truth
> > > > > unanswerable
> > > > >
> > > > > # model's answer
> > > > > <reasoning>
> > > > > The provided context and charts do not give any information about the speed at which a Gaussian distribution fits when using DANN features... (Omitted due to space limitations) The second chart shows Gaussian distributions for source and target data, but it does not mention DANN features or the speed of fitting. Therefore, the question cannot be answered based on the provided information.
> > > > > </reasoning>
> > > > > <answer>unanswerable</answer>
> > > > > ```
> > > > >
> > > > > ```markdown
> > > > > # InternVL3-38B, multiple choice question
> > > > >
> > > > > # question
> > > > > When comparing the performance of Euclidean and hyperbolic architectures on the PREFIX-30% task, which category of architecture can achieve higher accuracy in the training phase with fewer samples, and whose first-sentence embedding norm exhibits greater discriminative power?
> > > > > A. Euclidean gated recurrent unit + Euclidean feedforward network + Euclidean logistic regression
> > > > > B. Hyperbolic gated recurrent unit + Hyperbolic feedforward network + Hyperbolic/Euclidean hybrid logistic regression
> > > > > C. Hyperbolic recurrent neural network + Hyperbolic feedforward network + Hyperbolic logistic regression
> > > > > D. All architectures perform similarly in the low-sample phase, with differences emerging only later
> > > > >
> > > > > # Ground truth
> > > > > unanswerable
> > > > >
> > > > > # model's answer
> > > > > <reasoning>
> > > > > To determine which architecture performs better with fewer samples, we need to analyze the test accuracy and norm of the first sentence plots for each architecture during the initial phase of training ... (Omitted due to space limitations) 3. **Hyperbolic RNN + Hyperbolic FFNN + Hyperbolic MLR (Option C)**: - Test Accuracy: Similar to Option B, it shows a steeper initial increase in accuracy. - Norm of the First Sentence: The norm also increases more rapidly, indicating strong discriminative power.\n ... However, Option C (Hyperbolic RNN + Hyperbolic FFNN + Hyperbolic MLR) is more consistent with the context provided, which emphasizes hyperbolic architectures for better performance in hierarchical data like WordNet.
> > > > > </reasoning>
> > > > > <answer>C</answer>
> > > > > ```

---

> > > > > ### Author Response · Authors · 2025-11-28
> > > > > **Round 3 Response to Reviewer 6Moe (Part 2/2)**
> > > > >
> > > > > > Q3: An experiment where humans and 2–3 distinct model judges evaluate the "Open-Ended (sentence)" category to see whether rankings of models change under different evaluators.
> > > > >
> > > > > We appreciate the reviewer's suggestion to verify the robustness of our evaluation against different judges. To ensure our rankings are not artifacts of a specific judge model's bias, we conducted an extensive comparative study.
> > > > >
> > > > > We employed `Deepseek-chat` and `GPT-4o` as independent judges for the "Open-Ended (sentence)" category using the SEAT method. We compared their scoring distributions and resulting rankings against our original judge (Qwen3-32B) and human verification. Specifically, human verification was conducted by two experts on a sampled subset of 30 responses per model, whereas the automated judges were evaluated on the full benchmark.
> > > > >
> > > > > | As Judgement  | Qwen2.5-VL-7B | GLM-4.1V-9B | InternVL3-14B | Qwen2.5-VL-32B | InternVL3-38B |
> > > > > | ------------- | ------------- | ----------- | ------------- | -------------- | ------------- |
> > > > > | human         | 47.2          | 52.5        | 50.4          | 67.2           | 57.3          |
> > > > > | Qwen3-32B     | 49.5          | 68.8        | 56.3          | 72.7           | 60.0          |
> > > > > | Deepseek-chat | 44.8          | 64.7        | 59.1          | 70.9           | 61.0          |
> > > > > | GPT-4o        | 39.3          | 60.1        | 56.1          | 61.9           | 57.8          |
> > > > >
> > > > > The result reveals a high degree of consistency in model rankings across diverse judges, While absolute scores vary (e.g., GPT-4o is generally stricter, giving lower scores across the board), the relative ordering remains stable.
> > > > >
> > > > > We hope these responses can address your concerns. We sincerely appreciate the time and effort you have dedicated to reviewing our work and look forward to any further discussion.

---

### Official Review · Reviewer_yzXq · 2025-11-01

**Soundness:** 3
**Presentation:** 3
**Contribution:** 2
**Rating:** 4
**Confidence:** 5

**Summary:**

This paper identifies a significant weakness in Multimodal Large Language Models (MLLMs): while they excel at answering questions about single charts, their performance drastically declines when faced with complex, real-world tasks that require reasoning across multiple charts within a document. To address this evaluation gap, the authors introduce ChartNexus, a novel benchmark built from a large collection of real-world charts from domains like scientific papers and government reports. ChartNexus contains a curated set of questions designed to test multi-chart reasoning skills. A comprehensive evaluation of numerous state-of-the-art MLLMs on this benchmark reveals a substantial performance drop, uncovering key weaknesses in areas like cross-chart working memory and cross-modal reasoning. The study concludes that robust multi-chart understanding remains a major, unsolved challenge for MLLMs.

**Strengths:**

1. Identifies a Critical Research Gap: The paper successfully highlights a major disconnect between existing single-chart benchmarks and the complex needs of real-world document analysis, moving the field beyond an overemphasis on isolated chart understanding.

2. Novel and Rigorous Benchmark: The introduction of ChartNexus is a key contribution. Its strengths include: 1) Real-World Relevance:It is built from a large corpus of authentic charts from scientific, governmental, and industrial documents. 2) Systematic Design:It features a well-defined taxonomy of reasoning skills and difficulty levels, allowing for nuanced model diagnosis. 3) Complexity and Challenge:The benchmark is demonstrably challenging, causing significant performance drops even in top-tier models.

3. Comprehensive and Conclusive Evaluation:The large-scale evaluation of 23 diverse MLLMs provides strong, empirical evidence for the paper's central claim about the limitations of current models.

**Weaknesses:**

1. The overrepresentation of bar charts in the benchmark skews the overall evaluation. A benchmark for higher-difficulty tasks should prioritize more complex chart types to ensure a reliable and meaningful assessment of model capabilities.

2. A fine-grained, per-category performance breakdown is required. It is essential to identify if there are specific chart types that the model completely fails to process, revealing the true boundaries of its current abilities.

3. The insights presented are currently unsubstantiated, as they rely solely on textual description. Convincing validation requires dedicated ablation studies and visualizations to provide quantitative and tangible support for these claims.

4. The core distinction between multi-chart and single-chart reasoning must be clarified. The evaluation must go beyond reporting a performance gap and actively diagnose the underlying reasons for it, which is critical for understanding and advancing multi-chart reasoning.

**Questions:**

please refer to the weaknesses

---

> ### Author Response · Authors · 2025-11-24
> **Response to Reviewer yzXq (Part 1/2)**
>
> Thank you for your thorough and constructive feedback.
>
> > Q1: The overrepresentation of bar charts in the benchmark skews the overall evaluation. A benchmark for higher-difficulty tasks should prioritize more complex chart types to ensure a reliable and meaningful assessment of model capabilities.
>
> A1: We acknowledge the dominance of bar charts but respectfully emphasize that this distribution is a feature, not a bug, designed to reflect real-world usage. Our goal is to estimate model performance in realistic deployment scenarios (finance, policy, news), where bar and line charts constitute the vast majority of visualizations.
>
> However, to address the concern about "complex" charts, we performed a Stratified Performance Analysis as following. This revealed a counter-intuitive insight: models actually generalize well to rare 2D charts (e.g., Sankey, Tree maps). The true difficulty spike is not caused by "rare" types, but by 3D charts, where performance consistently collapses (e.g., Qwen2.5-VL-7B drops to 16.8%). This confirms ChartNexus effectively exposes model weaknesses without artificially inflating the distribution of rare chart types.
>
> | **models**     | **Bar** | **Line** | **Pie** | **Table** | **Scatter** | **Tree** | **Radar** | **Area** | **Other** | **Sunburst** | **Graph** | **Boxplot** | **Sankey** | **Heatmap** | **3D** | **Candlestick** | **Funnel** |
> | -------------- | ------- | -------- | ------- | --------- | ----------- | -------- | --------- | -------- | --------- | ------------ | --------- | ----------- | ---------- | ----------- | ------ | --------------- | ---------- |
> | GPT-4o         | 63.7    | 62.5     | 74.8    | 70.2      | 63.9        | 69.9     | 60        | 56.6     | 64.9      | 70.5         | 66.7      | 62          | 100.0      | 64.4        | 33.3   | 63.6            | 76.9       |
> | Qwen2.5-VL-7B  | 46.8    | 44.5     | 48.6    | 60.7      | 38.4        | 51.4     | 45        | 45.2     | 48.9      | 58.8         | 33.3      | 25          | 89.4       | 44.3        | 16.8   | 27.2            | 53.8       |
> | GLM-4.1V-9B    | 66.7    | 65.7     | 77.3    | 79.0      | 66.5        | 70.2     | 50        | 59.5     | 64.6      | 58.8         | 66.7      | 37.5        | 84.2       | 55.4        | 17.2   | 63.6            | 61.5       |
> | InternVL3-14B  | 58.3    | 58.7     | 79.1    | 80.1      | 36.8        | 50.6     | 55        | 63.6     | 40.2      | 66.7         | 53.3      | 23.5        | 89.4       | 20.0        | 20.0   | 45.4            | 61.5       |
> | Qwen2.5-VL-32B | 65.4    | 62.9     | 80.2    | 73.2      | 56.4        | 69.1     | 50        | 64.2     | 64.7      | 64.7         | 66.7      | 62.5        | 73.6       | 65.6        | 36.1   | 54.5            | 69.2       |
> | InternVL3-38B  | 63.1    | 63.8     | 65.2    | 77.2      | 81.6        | 66.6     | 65        | 66.6     | 75        | 64.7         | 46.7      | 68.6        | 100.0      | 85.0        | 33.3   | 63.6            | 84.6       |
>
> > Q2: A fine-grained, per-category performance breakdown is required.
>
> A2: We strictly agree with the reviewer that a holistic score can mask specific deficiencies. Identifying the "boundaries" of model capabilities is crucial for future improvements.
>
> In response, we have performed a **category-wise performance breakdown** across all 17 chart types present in our benchmark. The results reveal distinct performance tiers that map directly to the visual complexity of the tasks:
>
> 1. **Proficiency Zone (Standard 2D):** Models exhibit strong performance on standard 2D charts such as **Pie** (e.g., GPT-4o: 74.8%), **Table** (70.2%), and **Bar** (63.7%). This suggests that current MLLMs have achieved a high degree of robustness in handling common planar visualizations.
> 2. **The Capability Boundary (3D & Spatial):** As the reviewer hypothesized, we identified specific categories where models "fail to process" information effectively. The most striking boundary is observed in **3D charts**, where performance drops precipitously (e.g., GPT-4o: **33.3%**, Qwen2.5-VL-7B: **16.8%**).
> 3. **Consistency:** This "3D deficit" is consistent across all evaluated models, regardless of model size (from 7B to commercial APIs), highlighting a universal limitation in current vision encoders regarding spatial projection and depth perception in charts.

---

> > ### Author Response · Authors · 2025-11-24
> > **Response to Reviewer yzXq (Part 2/2)**
> >
> > > Q3: The insights rely on textual description; they need quantitative validation (e.g., ablation studies).
> >
> > A3: Thanks for your valuable advice. The core premise of ChartNexus is that real-world chart reasoning is not an isolated visual task but a multi-modal synergy problem.
> >
> > To rigorously validate this and clarify the distinction from single-chart reasoning, we performed a "Performance Without Context" ablation study. In this setting, we stripped all textual context (captions, article paragraphs) and fed only the chart images to the models. The results reveal a significant dependency on textual context, particularly for high-performing models in complex reasoning tasks.
> >
> > | **models**     | **Bool** | **Approximate value** | **Multi-Choice** | **Open-Ended (vocabulary)** | **Open-Ended (sentence)** |
> > | -------------- | -------- | --------------------- | ---------------- | --------------------------- | ------------------------- |
> > | Qwen2.5-VL-7B  | 31.7     | 34.5                  | 44.6             | 15.1                        | 43.2                      |
> > | GLM-4.1V-9B    | 41.7     | 40.6                  | 54.6             | 27.8                        | 55.5                      |
> > | InternVL3-14B  | 43.0     | 37.8                  | 53.4             | 25.2                        | 56.4                      |
> > | Qwen2.5-VL-32B | 52.3     | 42.7                  | 60.1             | 30.5                        | 68.1                      |
> > | InternVL3-38B  | 53.1     | 54.3                  | 51.5             | 21.2                        | 57.7                      |
> >
> > Comparing these results with the full-context performance reported in the our paper (Table 5), we observe critical "Context Gaps" that diagnose the underlying reasoning mechanism: The fact that models struggle significantly without text proves that the difficulty of ChartNexus stems from **cross-modal grounding** rather than just visual pattern recognition.
> >
> > We have integrated this analysis into `Section 4.3` and `Appendix G Additional Experimental Analysis` in the revision.
> >
> >  > Q4: The core distinction between multi-chart and single-chart reasoning must be clarified. The evaluation must go beyond reporting a performance gap and actively diagnose the underlying reasons for it, which is critical for understanding and advancing multi-chart reasoning.
> >
> > A4: We acknowledge that "multi-chart reasoning" can stem from two distinct structural settings: processing **Composite Images with Subplots as one image** or reasoning across **Multiple Discrete Images**.
> >
> > As noted in **Section 3.3**, $\approx 16.69\%$ of our charts are composite images containing multiple subplots. The remaining samples involve multiple distinct image files. We conducted a performance comparison between samples containing subplots (high visual density) and those involving only discrete single-chart images.
> >
> > | **Models**         | **Multi-Charts with Subplots** | **Multi-Charts w/o Subplots** | **Δ** |
> > | ------------------ | ------------------------------ | ----------------------------- | ----- |
> > | **InternVL3-38B**  | 64.2                           | 64.1                          | +0.1  |
> > | **Qwen2.5-VL-32B** | 60.3                           | 65.7                          | -5.4  |
> > | **GLM-4.1V-9B**    | 61.5                           | 68.2                          | -6.7  |
> > | **Qwen2.5-VL-7B**  | 46.5                           | 46.3                          | +0.2  |
> > | **InternVL3-14B**  | 37.7                           | 61.2                          | -23.5 |
> >
> > The results reveal a striking divergence that allows us to distinguish between "Visual Resolution Bottlenecks" and "Logical Reasoning Bottlenecks":
> >
> > Models (e.g., **InternVL3-14B**) exhibit a massive performance collapse (**-23.5%**) when dealing with subplots compared to discrete images. This diagnoses a critical limitation in the **visual encoder's resolution or capacity**: the model struggles to resolve the dense, packed details of multiple charts when they are compressed into a single image embedding, whereas it handles the same amount of information well when distributed across multiple image files.
> >
> > Larger models (e.g., **InternVL3-38B**) show virtually identical performance across both settings (64.2% vs 64.1%), indicating that scaling up the model (and likely its vision encoder) effectively overcomes this resolution bottleneck, allowing the model to focus on the logical reasoning task regardless of the visual layout.

---

### Official Review · Reviewer_QnKr · 2025-11-03

**Soundness:** 3
**Presentation:** 2
**Contribution:** 3
**Rating:** 6
**Confidence:** 3

**Summary:**

ChartNexus introduces a benchmark for evaluating multi-chart reasoning in multimodal large language models, focusing on realistic document-level understanding rather than isolated chart interpretation. The dataset combines real-world charts, accompanying text, and human-verified question–answer pairs to test models’ ability to integrate visual and contextual information. Compared to existing single-chart benchmarks, results show a substantial performance decline across all models, indicating that multi-hop and cross-modal reasoning remain unsolved challenges. While top commercial systems outperform open-source and chart-specialized models, they still struggle with numerical precision, contextual integration, and detecting unanswerable questions. Chain-of-thought prompting offers limited gains, effective mainly for numerical reasoning tasks.

**Strengths:**

1. Chart understanding is an important problem to work on an multimodal models generally struggle with this task.
2. The benchmark is well-constructed and covers many task types and chart types. The automatic annotation is balanced with high manual annotation agreement.
3. The evaluation covers over 20 models of many different types.

**Weaknesses:**

It is hard to tell what insights / takeaways are novel from this benchmark vs other ones. The comparison across benchmarks is good to see, but I would want to know what additional signal this benchmark provides. For instance charXiv already identified that commercial models generally outperform open-source models in real-world chart settings. I think this benchmark is good, but it is important to understand what trends it shows that we could not find otherwise.

**Questions:**

See above.

---

> ### Author Response · Authors · 2025-11-24
>
> > Q: It is hard to tell what insights/takeaways are novel compared to other benchmarks (beyond "Commercial > Open-source"). What additional signal does this benchmark provide?
>
> A: We share the reviewer's view that a benchmark must serve as a diagnostic tool, not just a leaderboard. Beyond reconfirming the dominance of commercial models, ChartNexus isolates three specific failure modes in current MLLMs that were invisible in prior benchmarks (Appendix G):
>
> 1. The "Context-Vision Gap": Unlike previous datasets, our experiments reveal that high performance relies heavily on cross-modal grounding. Open-source models struggle significantly more than commercial models to use textual context as a "semantic bridge" to resolve visual ambiguities.
> 2. The Resolution Bottleneck (Subplots): By comparing performance on Composite Images vs. Discrete Multi-Images, we identified that some smaller models (e.g., InternVL3-14B) suffer a significant performance drop on composite images due to visual encoder resolution limits, a finding unique to our dense-data setting.
> 3. The 2D-3D Collapse: Our stratified analysis identifies a universal "capability boundary." While models handle complex 2D charts well, we found a catastrophic performance collapse on 3D charts across all models (accuracy < 20–35%), signaling a fundamental lack of spatial/projective reasoning in current vision encoders.

---

### Official Review · Reviewer_mU4C · 2025-11-03

**Soundness:** 2
**Presentation:** 2
**Contribution:** 2
**Rating:** 2
**Confidence:** 4

**Summary:**

This paper introduces ChartNexus, a new benchmark dataset intended to evaluate the multi-chart and cross-modal (text + chart) reasoning capabilities of Multimodal Large Language Models (MLLMs). The authors argue that existing benchmarks are mostly restricted to single-chart queries, failing to capture the real-world scenarios. Empirical evaluation of 23 state-of-the-art MLLMs, including GPT-4o, exhibits a dramatic performance drop on this new benchmark.

**Strengths:**

1. A well-motivated problem, addressing a major concern in existing chart benchmarks that are mostly restricted to single-chart scenarios.
2. Collection of data from real-world sources.
3. Robust Human-in-the-Loop Annotation.
4. Multilingual extension.

**Weaknesses:**

1. The paper uses the Qwen3-32B model as the judge to evaluate the correctness of answers. However, the paper did not discuss the judge's accuracy against human scoring.

2. The abstract claims the benchmark comprises 6,793 real-world charts. However, Table 4 mentions only 3,198 charts. This is a major contradiction, making the dataset construction questionable in a datasets & benchmark-focused work.

3. While the paper reviewed various existing multi-chart benchmarks, it still lacks justification on how their contribution is not just an incremental contribution in comparison to the prior work.

4. Lack of details on what method was applied to measure inter-annotator agreement.

5. Lack of details on how the disagreement in open-ended QA is resolved.

**Questions:**

Address the weaknesses mentioned above.

---

> ### Author Response · Authors · 2025-11-24
>
> Thanks for your insightful feedback.
>
> > Q1: The paper uses the Qwen3-32B model as the judge to evaluate the correctness of answers. However, the paper did not discuss the judge's accuracy against human scoring.
>
> A1: We appreciate you raising this critical point regarding evaluation reliability. While Qwen3-32B is known for robust instruction following[1, 2], we recognize the need for task-specific validation. To address this, we conducted a Human-Model Alignment Study on a random sample of 200 examples.
>
> The results demonstrate a high consistency, with the model judge aligning with human experts in **93.5%** of cases. Our analysis shows that for objective tasks (Boolean, Multiple Choice), the judge's precision is near-perfect. Discrepancies were primarily confined to "Open-Ended" questions, where the model and humans occasionally differed on how to extract specific "sub-questions" for the SEAT evaluation metric. These results confirm that Qwen3-32B serves as a reliable proxy for human evaluation in this benchmark.
>
> [1] Whitehouse, Chenxi, et al. "J1: Incentivizing thinking in llm-as-a-judge via reinforcement learning." _arXiv preprint arXiv:2505.10320_ (2025).
>
> [2] Li, Qingquan, et al. "Evaluating Scoring Bias in LLM-as-a-Judge." _arXiv preprint arXiv:2506.22316_ (2025).
>
> > Q2: There is a contradiction between the abstract (6,793 charts) and Table 4 (3,198 charts).
>
> A2: We would like to clarify that the two numbers refer to different units of measurement, which reflects the specific nature of our multi-chart benchmark:
>
> - 3,198 refers to the number of distinct image files (`.png` files) stored in the dataset.
> - 6,793 refers to the total number of semantic atomic charts (including all subplots). As detailed in Section 3.3, approximately 16.69% of our image files are composite figures containing multiple subplots (averaging 4.78 subplots per composite image). We count these atomic subplots in the total because reasoning often requires extracting data from a specific subplot within a dense composite image, which is a key challenge in ChartNexus.
>
> > Q3: The paper lacks justification on how this benchmark is not just an incremental contribution compared to prior work.
>
> A3:
> - **Context-Dependency:** Unlike prior benchmarks (e.g., ChartQA, PlotQA) that treat charts as isolated images, ChartNexus is constructed to simulate real-world document querying. Our "Performance Without Context" ablation study (Appendix G.2) proves that SOTA models suffer performance drops when the textual context (captions, paragraphs) is removed. This confirms that our benchmark evaluates cross-modal grounding (aligning text with vision) rather than just visual extraction.
> - **Resolution & Complexity:** By including dense composite images, we introduce a "Visual Resolution" challenge absent in single-chart benchmarks, effectively diagnosing encoder capacity limits (as seen in the performance gap between 14B and 38B models).
>
> > Q4: Lack of details on the inter-annotator agreement measurement.
>
> A4: To ensure high-quality data, we employed a cross-validation strategy during the annotation phase:
>
> 1. We randomly sampled 20% of the annotated samples from the pool.
> 2. These samples were assigned to different annotators who blinded to the original labels. They were asked to annotate the data from scratch.
> 3. We measured agreement across both metadata and QA pairs. Specifically, we calculated the consistency for: (1) Chart type, (2) Presence of sub-charts, and (3) Number of sub-charts. (4) Answerability (whether the question can be answered based on the provided context).
>
> > Q5: Lack of details on how the disagreement in open-ended QA is resolved.
>
> A5: Regarding open-ended questions, we resolved potential ambiguity using the **SEAT (Semantic Evaluation for Answer Text)** [3] method. Instead of relying on rigid string matching, SEAT decomposes a holistic question into atomic sub-questions. We calculate an F1 score by comparing the answers to these sub-questions derived from the Ground Truth against those from the Model Prediction. This method allows for a robust, semantic-level agreement measurement even when phrasing differs.
>
> [3] Junnan Zhu, et al. 2025. [TableEval: A Real-World Benchmark for Complex, Multilingual, and Multi-Structured Table Question Answering](https://aclanthology.org/2025.emnlp-main.363/). In _Proceedings of the 2025 Conference on Empirical Methods in Natural Language Processing_, pages 7137–7157, Suzhou, China. Association for Computational Linguistics.

---

### Author Response · Authors · 2025-12-03
**Summary for the Area Chair**

We sincerely thank the Reviewers and the Area Chairs for their time and effort in reviewing our paper. The comprehensive feedback has significantly strengthened the rigor and depth of our work.

To address the reviewers' primary concerns and enhance the benchmark's value, we have incorporated the following major improvements and experiments into the revised manuscript (highlighted in blue):

---

**Ensuring Robust and Unbiased Evaluation**

**1. Validity of the Automated Judge:** *(Reviewer mU4C, 6Moe)*

We conducted a Human-Model Alignment Study. The results demonstrate high consistency, with the model judge aligning with human experts in 93.5% of cases overall.

- Structured Tasks: For Boolean, Multi-Choice, and Approximate Value questions (the majority of the benchmark), the judge achieves near-perfect alignment (98%-100%).
- Subjective Tasks: While minor misalignments exist in Open-Ended tasks, our multi-judge ablation (comparing Qwen3-32B, GPT-4o, and Deepseek) confirms that relative model rankings remain stable, proving that the judge introduces random noise rather than systematic bias.

**2. Establishment of Human Baseline:** *(Reviewer 6Moe)*

To quantify the "human-model gap," we recruited experts to establish a human performance baseline on stratified samples across all question types. We commit to conducting human baseline experiments on the full benchmark before the final version to serve as a gold-standard diagnostic set.

**3. Robustness to Prompt Language:** *(Reviewer 6Moe)*

We performed sensitivity analyses using both Chinese and English evaluation prompts for the SEAT metric. The results show consistent ranking trends, confirming the metric's robustness to prompt language variations.

---

**Novel Diagnostic Insights into Model Bottlenecks**

**4. Subplots vs. Discrete Images:** *(Reviewer mU4C, yzXq, 6Moe)*

We analyzed the distinct challenges posed by **Composite Subplots** versus **Discrete Images**. Our analysis reveals a "Visual Density Wall": models like InternVL3-14B exhibit a massive performance collapse (dropping from 61.2% to 37.7%) when processing subplots compared to discrete images. This diagnoses a critical bottleneck in visual encoder resolution/capacity when handling dense, packed information, a signal invisible in single-chart benchmarks.

**5. The Context-Vision Gap:** *(Reviewer yzXq)*

We conducted a "Performance Without Context" ablation, stripping all captions and contexts. The sharp performance drop confirms that ChartNexus tasks are not merely visual perception problems but require cross-modal grounding, where text acts as a semantic bridge. This validates the necessity of our document-level context design.

**6. Chart Type Analysis:** *(Reviewer yzXq)*

Our stratified analysis refutes the concern that "rare" types skew evaluation. We found that models handle rare 2D charts reasonably well, the true capability boundary lies in **3D charts**, where performance consistently collapses (e.g., Qwen2.5-VL-7B drops to 16.8%). This highlights a universal deficiency in spatial projection reasoning.

---

**Methodological Rigor and Sustainability**

**7. Justification for the "Boolean" Category:** *(Reviewer 6Moe)*

We provide empirical evidence justifying "Boolean" as a distinct category from "Multiple-Choice." Through a "Hallucination Analysis" on unanswerable questions, we found that models give in to **selection bias** in Multiple-Choice formats (hallucinating answers), whereas they demonstrate latent **fact-checking capabilities** in Boolean formats (correctly refusing to answer). This separation provides a crucial dimension for evaluating model safety and hallucination.

**8. Mitigation of Data Leakage:** *(Reviewer 6Moe)*

To ensure long-term reliability, we implemented a **Temporal Filter** strategy (prioritizing data from the last 1-6 months). Furthermore, we commit to open-sourcing our Automated Data Pipeline as a reproducible framework, allowing the community to generate fresh, contamination-free test sets from dynamic sources (e.g., World Bank, OECD) for future evaluations.

---

There are also many other insightful comments, which we address in detail in our full responses. We are pleased to note that our clarifications have resolved some of the concerns raised by Reviewer 6Moe. We thank the Area Chair and all reviewers again for all the efforts and valuable feedback.

---

### Meta-Review · Area_Chair_TxjB · 2026-01-07

**Summary:**

Paper was reviewed by four reviewers that initially provided the following ratings: 2 x rejects, 1x marginally above the acceptance threshold, 1x marginally below the acceptance threshold. Main concerns of reviewers centered around:

(1) Lack of novelty [mU4C] and/or insights / takeaways [QnKr, yzXq]
(2) Missing discussion of quality of LLM as a judge [mU4C, 6Moe]
(3) Misunderstanding on the size of the dataset [mU4C]
(4) Lack of details on how the disagreement in open-ended QA is resolved [mU4C]
(5) Potential overrepresentation of bar charts in the benchmark [yzXq]
(6) Lack of fine grained analysis and result breakdown [yzXq]
(7) Lack of clear motivation of multi-chart setting over the single-chart setting [yzXq]
(8) Unconvincing discussion on data leakage mitigation is unconvincing [6Moe]

Authors have attempted to address these concerns. However, [6Moe] while acknowledges some concerns being addressed, remains unconvinced overall. In general, while authors appear to adequately address a number of concerns, (1) remains a significant issue in the opinion of AC. Particularly, the arguments made with respect to insights, in response to [QnKr], are not very convincing . The "Context-Vision Gap"and the "Resolution Bottleneck" are generally well known limitations within the VLM community. The collapse of performance on 3D charts is interesting, but is a rather minor and limited insight. In light of this and the lack of more positive reviews, AC is recommending Rejection at this time.

**Reviewer Concerns:**

(1) Lack of novelty [mU4C] and/or insights / takeaways [QnKr, yzXq]
(2) Missing discussion of quality of LLM as a judge [mU4C, 6Moe]
(3) Misunderstanding on the size of the dataset [mU4C]
(4) Lack of details on how the disagreement in open-ended QA is resolved [mU4C]
(5) Potential overrepresentation of bar charts in the benchmark [yzXq]
(6) Lack of fine grained analysis and result breakdown [yzXq]
(7) Lack of clear motivation of multi-chart setting over the single-chart setting [yzXq]
(8) Unconvincing discussion on data leakage mitigation is unconvincing [6Moe]

**Reviewer Scores:**

While authors have addressed some concerns, they failed to address some of the most significant ones (like novelty and insights) that are shared by most reviewers. As a result, I do not think the reviewers would have updated their scores, at least to the point where the paper would be a strong contender for acceptance.  [6Moe] mentions explicitly that he will remain at the score of 4 -- marginally below the acceptance threshold. While this is an improvement over the original score of 2, it is short of pushing the paper over the acceptance threshold.

---

### Decision · Program_Chairs · 2026-01-26

Reject